## [Peer Review File · The EMBO Journal]

MYO5A-mediated stabilization promotes the acquisition of fusion competence in sealed autophagosomes

Akshaya Nambiar, Rene Martin, Kamakshi Tomar, Hans-Joachim Knölker, Sandhya Koushika, Subramaniam K., and Ravi Manjithaya

Corresponding author(s): Ravi Manjithaya (ravim@jncasr.ac.in)

Review Timeline:

Submission Date:	4th Sep 24
Editorial Decision:	21st Oct 24
Appeal Received:	13th Nov 24
Editorial Decision:	28th Nov 24
Appeal Received:	9th Sep 25
Editorial Decision:	8th Oct 25
Revision Received:	19th Nov 25
Accepted:	8th Dec 25

Editor: William Teale

Transaction Report:

Dear Ravi,

Thank you for submitting your study, "Myosin Va steadies autophagosomes to make them fusion-competent (EMBOJ-2024-118927), to EMBO Journal. Your manuscript was sent to three reviewers for evaluation; we have now received reports from all of them, which I enclose below. As you will see, despite some degree of enthusiasm for the topic and your general approach, your manuscript failed to garner the level of support from the referees for publication in EMBO Journal.

Your manuscript describes a role for MYO5A (a class V myosin) in the generation of a fusion-competent autophagosome-lysosome pose. Furthermore you show that MYO5A interacts directly with the autophagosome membrane, and that the described stabilising role is i) independent from the protein's motor domain and ii) necessary for pexophagy. You describe a particularly interesting hypothesis; however, at this stage we are unable to conclude that the data you present describe the hypothesised role with sufficient mechanistic depth.

Please note that at the EMBO Journal we subject to external review only those submissions that have a high chance of timely acceptance. I am sorry to disappoint you on this occasion and wish you success with the rapid publication of this dataset elsewhere.

Yours sincerely,

William

William Teale, PhD
Editor
The EMBO Journal
w.teale@embojournal.org

Referee #1:

The current study by Nambiar et al. builds on recent findings from the Manjithaya group, which investigated the role of general trafficking factors in autophagy. In this work, the authors focused on an unconventional class V myosin, MYO5A, and its role in regulating the fusion between autophagosomes and lysosomes. Using three biological models-yeast, mammalian cells, and *C. elegans*-they provide evidence supporting MYO5A's involvement in autophagy. The authors analyzed different domains of MYO5A and found that the CC-GTD domain (amino acids 923-1829) associates with LC3, a protein that marks mature autophagosomes. This interaction increases the number of stationary autophagosomes when directional movement is inhibited, suggesting a role in their consumption. While the proposed model for MYO5A's role in regulating autophagosome movement is promising, many results are descriptive, and the data do not fully support the suggested mechanisms. For instance, the study emphasizes the association between MYO5A and autophagosomes via LC3 interactions. The authors identified six predicted LIRs on the CC-GTD domain of MYO5A using the iLIR server. Still, they did not validate these interactions with LC3 or their role in inhibiting autophagosome movement. Additionally, the authors suggest that MYO5A interacts with lysosomes through LAMP1, which could be crucial for autophagosome-lysosome fusion. However, this potential interaction was not explored in the study. Overall, the hypothesis regarding MYO5A's role in forming competent SNARE and HOPS fusion complexes requires significant experimental validation, lacking in this research.

Additional Comments:

Figure 1: The authors should complement their western blot results with immunofluorescence (IF) experiments, as mentioned in Fig. 1a. This aspect of the study needs further exploration to clarify the nature of the intermediates that accumulate in the absence of Myo2.

Figure 2: In line 147, it states: "MYO5AKO cells, upon PBP treatment, were comparatively larger in size..."-however, the knockout cells were not treated with PBP, at least as presented. When the authors claim to inhibit autophagic flux (panels h-i), they should measure this flux with and without BabA1 (which is missing for the starvation condition) and quantify the results accordingly.

Figure 5: The protection assay should be conducted in MYO5A knockout (KO) cells instead of relying on siRNA.

Figure 7b: The choice of colors is misleading; cells expressing GFP-MYO5A (which should appear green) are represented using mCh-LC3. Additionally, the localization of all three proteins needs to be determined.

Referee #2:

Roles of unconventional myosins in autophagy are less characterized. Here, Nambiar et al., reveal that class V myosin, MYO5A

is a critical regulator of autophagy by utilizing several models, including yeast, worms, and mammalian cells. They demonstrate that inhibition of MYO5A impairs autophagosome-lysosome fusion. Mechanistically, MYO5A is required for the recruitment of several autophagy components which is essential for the fusion on autophagosomes. They showed that MYO5A interacts with autophagosomes through its coiled-coiled and globular tail domains. Live imaging analysis indicates that MYO5 is critical to increase stationary autophagosomes. Although the data shown could be potentially interesting, the manuscript is very descriptive and lack clear mechanistic insights. The specific comments are listed below.

Major

1. It is unclear which functions of MYO5 is critical to regulate autophagosome-lysosome fusion. In Figure 5e and Figure 8, it is difficult to understand the mechanism in which the regulation of directional switch of autophagosome by MYO5A contributes to the interaction between LC3 and VPS33A or STX17. In addition, the authors need to elucidate whether the re-introduction of FL and CC-GTD constructs in MYO5A KO cells also improves the decreased interactions of STX17 and VPS33A with LC3. Furthermore, in Figure 2h and 2i, Figure 7, and Figure 8, the author should confirm the autophagy flux carefully by comparing LC3-II in the presence and absence of BafA1.
2. It is important to show if the mutations in LIR of MYO5A affect the autophagosome-lysosome fusion and the pool of stationary autophagosomes.
3. In the introduction part, the authors mention that MYO5A mutation causes the defect in pexophagy. Is MYO5A also required for the autophagosome-lysosome fusion during several selective autophagy such as mitophagy, lipophagy and ERphagy?
4. Related to Figure 4i, is the increased Igg-1 puncta in the Proximal process in hum-2 suppressed by expression of exogeneous HUM-2 in *C. elegans*? Similarly, the other phenotypes such as no. of embryos laid and percentage of worm survival in hum-2 in Suppl Figure 3h and 3i are also rescued by expression of exogeneous HUM-2 ?
5. The author should carefully make the Synopsis. They show the VAMP8 on the lysosome which interacts with SNAP29 in Synopsis but there is no data about the VAMP8 in this paper.
6. The author needs to quantify the efficiency of the entrapped lysosome to autophagosome in Figure 2e, perinuclear clustering lysosome in Figure 3a, and the colocalization of the VPS33A with the autophagosomes in Figure 5d. Figure 3 may be better to move to the Supp. Instead, Supp Figure 3h and 3i would be better to show as the main Figure. The author should indicate the statistical difference in Figure 3i.
7. In line 138, the author mentioned that PBP induces the slowing movement of the MYO5A on the action cytoskeleton. It is more informative if you can show the phenotypes induced by PBP treatment such as the accumulation of the autophagosome may be canceled by the addition of FL and CC-GTD constructs.

Minor

1. Are there any reports that other MYOs such as MYO6 control the directional switch of autophagosome? The author needs to mention about this.
2. In Figure 1C, it might be confused to say "Myo2-12 showed significantly higher level of free GFP at PT compared to NPT conditions." You can rewrite correctly for example showing the time.
3. There are many discrepancies among the sentence and the figure number. Do Supp. Fig. 1e in line 119, Supp. Fig. 1f in line 121, and Supp. Fig. 1e in line 122 mean Supp. Fig. 1c, 1e, and 1d, respectively? Can you confirm the line 248 and lines 714-720? Does Supp. Figure 4 indicate Supp. Figure 3?

Referee #3:

Review of Nambiar et al.

Using three very different experimental systems this manuscript argues for a role for a type V myosin at a late stage of autophagy, after the autophagosome has formed a closed structure and come into proximity to the lysosome but prior to fusion. There are numerous technical issues, but my main concern is with the proposed model, as shown in the synopsis. The left panel of the synopsis implies that the motor domain of the myosin is acting as a brake for the autophagosome by binding to an actin fiber, however Fig 7 shows that the motor domain is not involved in autophagosome immobilization, rather the CC-GTD domains serve this role by interacting with a variety of components on the autophagosome and tethering it to the lysosome. There are other internal contradictions as well. In Fig 2 an inhibitor, PBP, is used which acts on the motor domain and this is shown to block fusion with lysosomes. So, is the motor domain required or not? This issue must be resolved. My overall impression is that

by combining studies on three different system, they have not delved deeply enough into any of them.

Specific issues:

Fig 1. They make use of a number of temperature-sensitive mutations in the yeast MYO2 gene. Where are these mutations within the myosin sequence? They compare GFP-Atg8 cleavage at the permissive temperature to the same strain at the restrictive temperature, but this ignores temperature effects on the pathway. It is more typical to compare mutant to WT at each temperature. They incubate the mutants for 4 hrs at the restrictive temperature. This may be too long for these conditionally lethal mutants, resulting in severe effects on cell metabolism and/or integrity. In panel C the mutant exhibits free GFP at 1 hr that then goes down. This is very unusual behavior. The WT in panel C shows much less cleavage than in panel B, why? In panel F there are two different cleavage products, why? In supplementary fig 1 they shift the mutants to 36C for 15 hrs. This is far too long for a conditionally lethal mutant.

Fig 2. In panel B the inhibitor results in a large increase in the number of autophagosomes, but the KOs do not, why? In panel D, KO2 results in a much larger increase in the area of LC3 puncta than KO1 or the inhibitor, why? On line 147 they say "Interestingly, we also observed that autophagosomes formed in MYO5A KO cells, upon PBP treatment, were comparatively larger in size". This seems to imply that they are adding the inhibitor to the KO, which makes no sense. In line 176 they mix the order of Lamp1 and Rab7.

Fig 3 is mostly negative data in that it shows a variety of parameters that are not affected by MYO5 KO. As such this figure could be moved to the supplement.

Fig 4. Where is the hum-2 (ok596) mutation within the myosin sequence?

Fig 5. Was the IP performed after detergent solubilization or were they trying to pull down intact organelles? Explain. In panel E, why was STX17 reduced in the KO IP but not Vps33A or SNAP29? What does this mean? Does Myo5 bind to any of these components in vitro using purified proteins or just in whole cell lysates? The model in the Synopsis does not indicate interaction of Myo5 GTD with any specific component, but the text suggests involvement of LIR domains. This is very confusing.

Fig 6. When expressing the various Myo5 constructs were the cells deleted for the endogenous copy? If not, heterodimer formation could yield confusing results.

Fig 7. The apparent complementation of the KO by the CC-GTD construct lacking the motor domain appears to be in conflict with the inhibitor data in Fig 2 and with the model in the Synopsis. This needs to be resolved as it relates to the central point of the paper.

** As a service to authors, EMBO Press provides authors with the possibility to transfer a manuscript that one journal cannot offer to publish to another EMBO publication or the open access journal Life Science Alliance launched in partnership between EMBO Press, Rockefeller University Press and Cold Spring Harbor Laboratory Press. The full manuscript and if applicable, reviewers' reports, are automatically sent to the receiving journal to allow for fast handling and a prompt decision on your manuscript. For more details of this service, and to transfer your manuscript please click on Link Not Available. **

Referee #1

The current study by Nambiar et al. builds on recent findings from the Manjithaya group, which investigated the role of general trafficking factors in autophagy. In this work, the authors focused on an unconventional class V myosin, MYO5A, and its role in regulating the fusion between autophagosomes and lysosomes. Using three biological models-yeast, mammalian cells, and *C. elegans*. They provide evidence supporting MYO5A's involvement in autophagy.

The authors analysed different domains of MYO5A and found that the CC-GTD domain (amino acids 923-1829) associates with LC3, a protein that marks mature autophagosomes. This interaction increases the number of stationary autophagosomes when directional movement is inhibited, suggesting a role in their consumption.

- While the proposed model for MYO5A's role in regulating autophagosome movement is promising, many results are descriptive, and the data do not fully support the suggested mechanisms. For instance, the study emphasizes the association between MYO5A and autophagosomes via LC3 interactions. The authors identified six predicted LIRs on the CC-GTD domain of MYO5A using the iLIR server. Still, they did not validate these interactions with LC3 or their role in inhibiting autophagosome movement.
- We thank the reviewer for this input. We have shown the interaction between autophagosome and MYO5A in multiple ways including co-localisation analysis and immuno-pulldown experiments (Fig 6, Fig 5e and Fig 5g). **Additionally, to further validate these interactions, we would like to include proximity ligation assay data to show direct interaction between MYO5A and LC3 puncta.**
- Our results suggest that this interaction is mediated via the CC-GTD domain of the MYO5A protein (Fig 6 and Fig 7) as re-introduction of mEmerald CC-GTD plasmid construct was enough to rescue the decrease in co-localisation observed between lysosomal compartments marked with LAMP1 and autophagosomal compartments marked with mCh-LC3 in *MYO5A* KO cells. **This rescue in the LC3-LAMP1 interaction has also been validated using proximity ligation assays (PLA, Fig 7d and 7e).**
- We speculated that this interaction might be mediated via the LC3 Interacting Region (LIR) motifs which we mapped to the CC-GTD domain of the protein. As suggested by the reviewer, we do need to validate these interactions using immunofluorescence and immuno-pulldown assays after the creation of LIR mutations in the MYO5A protein. Since we did anticipate this comment, we have already in process of generating these LIR mutants of MYO5A using site-directed mutagenesis approach. We will be able to perform the LIR experiments in a time-period of **2 months**.

Experiments – Creation of LIR mutations and experimental validations

Time required – 2 months

- **Additionally, the authors suggest that MYO5A interacts with lysosomes through LAMP1, which could be crucial for autophagosome-lysosome fusion. However, this potential interaction was not explored in the study.**
- Our study and experimentations did focus on both autophagy and endo-lysosomal pathways. To understand if loss of MYO5A affects the endo-lysosomal pathway, we assessed for the

lysosomal parameters including its morphology, function, pH and motion properties and did not observe any significant differences between the *MYO5A* KO cells and WT. We also assessed for the distribution of lysosomes (marked with CTNS::mCh) in the PLM neurons of *C. elegans* and found no significant differences between the WT and *hum-2(ok596)* mutant worms.

- These results suggest that absence of MYO5A by both genetic and pharmacological means does not perturb the endo-lysosomal pathway. Therefore, we speculate that the inefficient autophagosome-lysosome fusions observed in the absence of MYO5A does not stem from the lysosomes being dysfunctional or due to perturbations in interaction of MYO5A with LAMP1 and interaction of MYO5A with LAMP1 seems to be dispensable for fusion. Although, we will now modify our results and discussion section to amplify this fact to avoid confusion.

- **Overall, the hypothesis regarding MYO5A's role in forming competent SNARE and HOPS fusion complexes requires significant experimental validation, lacking in this research.**

We thank the reviewer for this comment. We understand that proposed role of MYO5A in the formation of competent SNARE and HOPS fusion complexes does require further experimental validations.

Experiments – While we have pull immunoprecipitated MyoVa and shown interaction with SNARE and HOPS complexes, we will carry out reciprocal IPs with STX17 and VPS33A (HOPS subunit) to characterise these interactions. We will particularly probe for SNAP29, Rab7, LAMP1, LC3 and VAMP8.

Time required – 2 months

- **The authors should complement their western blot results with immunofluorescence (IF) experiments, as mentioned in Fig. 1a. This aspect of the study needs further exploration to clarify the nature of the intermediates that accumulate in the absence of Myo2.**

We will perform the complementary experiments (**fluorescence microscopy**) as mentioned in the guidelines for the GFP-Atg8 assay (Klionsky et al. 2021) (at NPT for the *myo2* mutants).

Experiment – Fluorescence microscopy in *ts* mutant strains of Myo2

Time required – 2 weeks

- **In line 147, it states: "MYO5A KO cells, upon PBP treatment, were comparatively larger in size..."-however, the knockout cells were not treated with PBP, at least as presented. When the authors claim to inhibit autophagic flux (panels h-i), they should measure this flux with and without BafA1 (which is missing for the starvation condition) and quantify the results accordingly.**

- We thank the reviewer for this comment and bringing up this confusion. As the reviewer rightly noticed, *MYO5A* KO cells were not treated with PBP, only WT cells were treated with PBP. The sentence will be corrected to “Interestingly, we also observed that autophagosomes formed in *MYO5A* KO cells, **and** upon PBP treatment, were comparatively larger in size (Fig. 2d).”

- Also as pointed out, we will measure the autophagic flux with a combinatorial treatment of BafA1 and starvation and quantify the results accordingly.

Experiment –Western blotting (+BafA1 +EBSS)

Time required – 2 weeks

- **The protection assay should be conducted in *MYO5A* knockout (KO) cells instead of relying on siRNA.**

We thank the reviewer for this comment. We obtained our results in KD cells prior to obtaining the KO cells. In the KO cells, the block in autophagy flux, increase in size of the autophagosomes and related observation match with that of the KD cells. We assume that the results pertaining to protease protection assay would be similar in *MYO5A* KO cells as we do achieve ~85% knock-down on siRNA mediated knock-down of *MYO5A*.

- **The choice of colours is misleading; cells expressing GFP-MYO5A (which should appear green) are represented using mCh-LC3. Additionally, the localization of all three proteins needs to be determined.**
- We used these pseudo colours for representation as green and magenta are **colour-blind friendly** and the colocalization events between them appear as white which has been quantified in the figure (Fig 7b). We can surely include in the legends, results and material methods to highlight this change.
- We thank the reviewer for the comment on additional (triple) co-localisation analysis which we will perform.

Analysis – Triple colocalization between LC3, LAMP1 and MYO5A

Time required – 1 week

Referee #2:

Roles of unconventional myosins in autophagy are less characterized. Here, Nambiar et al., reveal that class V myosin, MYO5A is a critical regulator of autophagy by utilizing several models, including yeast, worms, and mammalian cells. They demonstrate that inhibition of MYO5A impairs autophagosome-lysosome fusion. Mechanistically, MYO5A is required for the recruitment of several autophagy components which is essential for the fusion on autophagosomes. They showed that MYO5A interacts with autophagosomes through its coiled-coiled and globular tail domains. Live imaging analysis indicates that MYO5A is critical to increase stationary autophagosomes. Although the data shown could be potentially interesting, the manuscript is very descriptive and lack clear mechanistic insights. The specific comments are listed below.

- **It is unclear which functions of MYO5A is critical to regulate autophagosome-lysosome fusion. In Figure 5e and Figure 8, it is difficult to understand the mechanism in which the regulation of directional switch of autophagosome by MYO5A contributes to the interaction between LC3 and VPS33A or STX17. In addition, the authors need to elucidate whether the re-introduction of FL and CC-GTD constructs in MYO5A KO cells also improves the decreased interactions of STX17 and VPS33A with LC3. Furthermore, in Figure 2h and 2i, Figure 7, and Figure 8, the author should confirm the autophagy flux carefully by comparing LC3-II in the presence and absence of BafA1.**
- Thank you for these insightful suggestions. We agree that it is worthwhile to examine the **restoration of the interaction between LC3 and components of the fusion machinery upon reintroduction of the FL and CC-GTD constructs of MYO5A.** To address this, we plan to perform rescue experiments using immunofluorescence (IF), immunoprecipitation (IP), and proximity ligation assays (PLA) to more accurately assess these interactions.

- Furthermore, as suggested, we will confirm autophagic flux in Figures 2h, 2i, 7, and 8 by comparing LC3-II levels in the presence and absence of BafA1 to ensure accurate assessments. Thank you again for your valuable feedback, which will help enhance the rigor of our study.

Experiments – Rescue of decreased interaction between LC3 and fusion machinery components by IF, IP and PLA

Time required – 2 months

- **It is important to show if the mutations in LIR of MYO5A affect the autophagosome-lysosome fusion and the pool of stationary autophagosomes.**

We thank the reviewer for this comment. We have already **constructed the LIR mutants** of MYO5A and now will be validating its importance in mediating the interaction of MYO5A with LC3 by performing **immunofluorescence, immuno-pulldown and live-imaging assays.**

Experiments – Creation of LIR mutations and experimental validations

Time required – 2 months

- **In the introduction part, the authors mention that MYO5A mutation causes the defect pexophagy. Is MYO5A also required for the autophagosome-lysosome fusion during several selective autophagy such as mitophagy, lipophagy and ERphagy?**

We show that general autophagy is affected in both yeast and mammalian cells. Our initial screens were based on the pexophagy assay and that is how we identified Myo2 as a player in autophagy. As it directly affects fusion, we expect all macroautophagy related pathways will be affected by loss of Myo2 function as the autophagosome fusion itself is perturbed.

- **Related to Figure 4i, is the increased lgg-1 puncta in the Proximal process in hum-2 suppressed by expression of exogenous HUM-2 in C. elegans? Similarly, the other phenotypes such as no. of embryos laid and percentage of worm survival in hum-2 in Supp-1 Figure 3h and 3i are also rescued by expression of exogenous HUM-2?**

We thank the reviewer for this insightful comment. We did not perform a rescue experiment on autophagic defects or life history traits through exogenous expression of hum-2. However, we conducted preliminary assessments of **brooding capacity** and embryonic lethality in the **heterozygous hum-2(ok596) mutant**, in which one copy of the mutant allele was replaced with the wild-type allele by crossing with wild-type worms. Our findings indicate that the reduced brooding percentage **was restored** in the heterozygous mutant worms (see figure below). Further experiments could be performed in these heterozygous mutants to validate and extend our findings.

Figure: Brood size indicates the ability of *C. elegans* to produce offspring. The graph represents the number of embryos laid by an adult WT (N2) and *hum-2(ok596)/+* heterozygous mutant worms and is plotted and represented as mean \pm SEM between the genotypes (N=1, n=10 worms).

Experiments – Rescue of *C. elegans* phenotypes in the heterozygous *hum-2(ok596)* mutant worms

Time required – 4 months

The author should carefully make the Synopsis. They show the VAMP8 on the lysosome which interacts with SNAP29 in Synopsis but there is no data about the VAMP8 in this paper.

We thank the reviewer for this comment. We had performed colocalisation-based experiments between the lysosomal marker LAMP1 and VAMP8 (Flag-VAMP8) and **did not observe any significant differences in the percentage colocalization between the two compartments** (Graph below). This data was not included in the previous manuscript, we will include this as a part in the revised version to substantiate the synopsis. This data will be included as a supplementary figure.

Figure: Wild-type (WT) and *MYO5A* knockout (KO) cells were transfected LAMP1-GFP and Flag-VAMP8. The graph represents the percentage of co-localization events between different stage-specific markers and LC3B quantified using the “Colocalization” plug-in of ImageJ in WT and *MYO5A* KO cells. The percentage of co-localization has been plotted as mean \pm SEM from three independent experiments (N=3, n=60 cells). Statistical significance was assessed by Unpaired student’s t-test (ns-non-significant).

- **The author needs to quantify the efficiency of the entrapped lysosome to autophagosome in Figure 2e, perinuclear clustering lysosome in Figure 3a, and the colocalization of the VPS33A with the autophagosomes in Figure 5d. Figure 3 may be better to move to the Supp. Instead, Supp Figure 3h and 3i would be better to show as the main Figure. The author should indicate the statistical difference in Figure 3i.**

Analysis which can be included:

Timeline – 1 month

- **Colocalization analysis** between mCh-LC3 and LAMP1-GFP to quantify the efficiency of autophagosome capture by lysosomes in Fig 2e.
- Quantification of **perinuclear clustering** in Fig 3a can be performed using **Radial plot** plugin of FIJI ImageJ which quantifies the distribution of the puncta inside a cell from the nucleus.
- **Colocalization analysis** on the live-cell experiments between **GFP-VPS33A** and **mCh-LC3** in Fig 5d.

- **In line 138, the author mentioned that PBP induces the slowing movement of the MYO5A on the actin cytoskeleton. It is more informative if you can show the phenotypes induced by PBP treatment such as the accumulation of the autophagosome may be cancelled by the addition of FL and CC-GTD constructs.**

PBP was employed as a chemical inhibitor to impede MYO5A movement along the actin cytoskeleton (Fedorov et al. 2009). Although we have not directly tested whether autophagosome accumulation observed with PBP treatment could be rescued by the addition of FL or CC-GTD constructs of MYO5A, we hypothesize that **PBP would inhibit FL MYO5A activity** due to its interaction with the motor domain. Nonetheless, it would be worthwhile to **investigate if introducing the CC-GTD domain of MYO5A, which lacks the motor domain, could restore autophagosome accumulation**, as this construct should theoretically rescue the phenotype.

Experiments – Rescue of autophagic flux on PBP treatment by reintroduction of CC-GTD domain (IF and western blotting)

Time required – 1 month

Referee #3:

- **Review of Nambiar et al. Using three very different experimental systems this manuscript argues for a role for a type V myosin at a late stage of autophagy, after the autophagosome has formed a closed structure and come into proximity to the lysosome but prior to fusion. There are numerous technical issues, but my main concern is with the proposed model, as shown in the synopsis. The left panel of the synopsis implies that the motor domain of the myosin is acting as a brake for the autophagosome by binding to an actin fibre, however Fig 7 shows that the motor domain is not involved in autophagosome immobilization, rather the CC-GTD domains serve this role by interacting with a variety of components on the autophagosome and tethering it to the lysosome.**
- Our experimental evidence indicates that the CC-GTD domain of MYO5A is crucial for mediating interactions with autophagosomes. As shown in Figures 6 and 8, **motor domain overexpression inhibits autophagosome-lysosome fusion**. Live imaging analysis further reveals that reintroducing the motor domain in MYO5A KO cells increases the frequency of directional changes in autophagosomes. That is why in the synopsis we show the motor domain of the protein being tethered on the actin cytoskeleton and the tail region (CC-GTD) interacting with the autophagosomes. We understand the confusion and will be updating both the synopsis and discussion part of the manuscript to suggest that **although CC-GTD is important for the interaction of MYO5A with the autophagosomes, the motor domain is not indispensable in the context of autophagic flux**. We have detailed below the experiments which can be performed to elucidate the same which will further strengthen the mechanistic aspects of the manuscript.

Experiments

To elucidate the importance of motor domain: Performing autophagy related experiments wherein the specific motor domain mutants, G440A which increases the interaction of MYO5A with the actin cytoskeleton and S217A, which reduces the duty ratio of MYO5A (Evans et al. 2014) will be used to delineate the contribution of motor domain of MYO5A in autophagic flux.

Timeline – 3 months

- **The apparent complementation of the KO by the CC-GTD construct lacking the motor domain appears to be in conflict with the inhibitor data in Fig 2 and with the model in the Synopsis. This needs to be resolved as it relates to the central point of the paper. There are other internal contradictions as well. In Fig 2 an inhibitor, PBP, is used which acts on the motor domain and this is shown to block fusion with lysosomes. So, is the motor domain required or not? This issue must be resolved. My overall impression is that by combining studies on three different system, they have not delved deeply enough into any of them.**

PBP was used as a chemical inhibitor to impede MYO5A movement along the actin cytoskeleton (Fedorov et al. 2009). To elucidate the role of the MYO5A motor domain in autophagic flux, we plan to conduct rescue experiments where the PBP-induced inhibition of autophagic flux would be restored by introducing FL and CC-GTD constructs of MYO5A. We hypothesize that PBP would specifically inhibit FL MYO5A activity due to its interaction with the motor domain. Furthermore, it would be valuable to investigate whether the introduction of the CC-GTD domain of MYO5A, which lacks the motor domain, could restore autophagosome accumulation, as this construct may potentially rescue the phenotype.

Experiments – Rescue of autophagic flux on PBP treatment by reintroduction of CC-GTD domain (IF and western blotting)

Time required – 1 month

- **They make use of a number of temperature-sensitive mutations in the yeast MYO2 gene. Where are these mutations within the myosin sequence?**

The mutations in the Myo2 sequence all map to the tail region of the protein. All the *ts* mutant constructs of Myo2 were sourced from Anthony Bretscher's lab (Schott et al. 1999).

myo2-16 (M1212T, L1471S, D1497V)

myo2-12 (H1373R, Q1441L, D1457V, S1512T)

myo2-17 (K1285R, Y1287N, L1436S)

myo2-18 (Y1161F, N1171Y, L1413S, I1453V, I1498M)

Figure: Schematic representation of different domains of Myo2 protein and highlighted box indicates tail domain (1087-1574 aa).

- **They compare GFP-Atg8 cleavage at the permissive temperature to the same strain at the restrictive temperature, but this ignores temperature effects on the pathway.**

We have included the WT control at both temperatures.

- **It is more typical to compare mutant to WT at each temperature.**

We now provide graphs comparing mutant with the WT at non-permissive temperatures.

Figure: Yeast cells expressing GFP-Atg8 were starved in SD-N medium. Starvation induces autophagosomes targeted to the vacuole, and GFP, which is highly protease-resistant, accumulates in the vacuole and is seen as free GFP release upon immunoblotting. The graph depicts the intensity ratio of free GFP against GFP-Atg8 of WT (*MYO2*) and *ts* mutant strains of Myo2 (as indicated) at NPT (non-permissive temperature). The graph represents the mean of three independent experiments (error bars: SEM). Statistical significance was assessed by one-way ANOVA with Bonferroni post-hoc analysis (** $p < 0.01$, *** $p < 0.001$)

- They incubate the mutants for 4 hrs at the restrictive temperature. This may be too long for these conditionally lethal mutants, resulting in severe effects on cell metabolism and/or integrity.

We will perform cell viability assays to determine the health of the cells for the duration of assay.

- In panel C the mutant exhibits free GFP at 1 hr that then goes down. This is very unusual behaviour.

This appears to be technical issue as the delay in inactivation of the mutant Myo2 while shifting to NPT may result in initial appearance of free GFP.

- The WT in panel C show much less cleavage than in panel B, why?

Panel C represents the GFP-Atg8 processing assay in *myo2-12 ts* mutant allele of Myo2 while panel B represents the WT. The differences in the GFP processing are stemming from the presence of different *ts* mutant alleles integration.

- In panel F there are two different cleavage products, why? In supplementary fig 1 they shift the mutants to 36C for 15 hrs. This is far too long for a conditionally lethal mutant.

We will perform cell viability assays to determine the health of the cells for the duration of assay.

Experiments – Cell viability assays of *ts* mutants of *Myo2* in permissive and non-permissive temperatures.

Timeline – 1 month

- **Where is the *hum-2* (*ok596*) mutation within the myosin sequence?**

The hypo-morphic mutant strain *hum-2(ok596)* was generated as part of the *C. elegans* Gene Knockout Project at the Oklahoma Medical Research Foundation, a member of the International *C. elegans* Gene Knockout Consortium wherein 2.1kb of the protein has been deleted (<https://cgc.umn.edu/gene/hum-2>) and we also validated the same using PCR.

Figure. a. SnapGene map of the *hum-2* gene, showing the location of primers used to differentiate between the wild-type (WT) and deletion mutant *hum-2(ok596)* alleles. The primers are positioned to produce distinct amplicon sizes for each genotype. **b.** Representative agarose gel image of PCR products. WT samples show a 3145 bp band, while *hum-2(ok596)* mutant samples yield a 1045 bp band, confirming successful distinction between the WT and deletion mutant allele.

- **Was the IP performed after detergent solubilization or were they trying to pull down intact organelles? Explain.**

We thank the reviewer for this comment. Immuno-pulldown experiments were performed after detergent solubilisation. The reason for carrying out IP after detergent solubilization is to avoid indirect interactions that may appear due to carry over of membrane fragments harbouring proteins.

- **In panel E, why was STX17 reduced in the KO IP but not Vps33A or SNAP29? What does this mean?**
- We have shown through immuno-precipitation that MYO5A interacts with STX17 (fig 5g). We speculate that the loading of STX17 onto the mature autophagosomes prior to fusion might require the autophagosomes to be in stable configuration as proposed in our model.
- We will further validate this hypothesis by performing experiments aimed at restoration of the interaction between LC3 and components of the fusion machinery upon reintroduction of the FL and CC-GTD constructs of MYO5A. To address this, we plan to perform rescue experiments using immunofluorescence (IF), immunoprecipitation (IP), and proximity ligation assays (PLA) to more accurately assess these interactions.

Experiments – Rescue of decreased interaction between LC3 and fusion machinery components by IF, IP and PLA

Time required – 2 months

- **Does Myo5 bind to any of these components in vitro using purified proteins or just in whole cell lysates? The model in the Synopsis does not indicate interaction of Myo5 GTD with any specific component, but the text suggests involvement of LIR domains. This is very confusing.**

We thank the reviewer for this comment. **We have already constructed the LIR mutants of MYO5A** and now will be validating its importance in mediating the interaction of MYO5A with LC3 by performing **immunofluorescence, immuno-pulldown and live-imaging assays**. We will further update our synopsis and discussion section to reflect the same.

Experiments – Creation of LIR mutations and experimental validations

Time required – 2 months

- **When expressing the various Myo5 constructs were the cells deleted for the endogenous copy? If not, heterodimer formation could yield confusing results.**

Thank you for this insightful comment. We confirm that MYO5A constructs were expressed in *MYO5A* KO cells, ensuring there is no endogenous copy present (Fig 7 and Fig 8). This eliminates the possibility of heterodimer formation and allows for clearer interpretation of our results.

References:

- Evans, Richard D., Christopher Robinson, Deborah A. Briggs, David J. Tooth, Jose S. Ramalho, Marta Cantero, Lluís Montoliu, Shyamal Patel, Elena V. Sviderskaya, and Alistair N. Hume. 2014. “Myosin-Va and Dynamic Actin Oppose Microtubules to Drive Long-Range Organelle Transport.” *Current Biology* 24 (15): 1743–50.
- Fedorov, Roman, Markus Böhl, Georgios Tsiavaliaris, Falk K. Hartmann, Manuel H. Taft, Petra Baruch, Bernhard Brenner, René Martin, Hans-Joachim Knölker, and Herwig O. Gutzeit. 2009. “The Mechanism of Pentabromopseudilin Inhibition of Myosin Motor Activity.” *Nature Structural & Molecular Biology* 16 (1): 80–88.
- Klionsky, Daniel J., Amal Kamal Abdel-Aziz, Sara Abdelfatah, Mahmoud Abdellatif, Asghar Abdoli, Steffen Abel, Hagai Abeliovich, et al. 2021. “Guidelines for the Use and Interpretation of Assays for Monitoring Autophagy (4th Edition) ¹.” *Autophagy* 17 (1): 1–382. <https://doi.org/10.1080/15548627.2020.1797280>.
- Schott, Daniel, Jackson Ho, David Pruyne, and Anthony Bretscher. 1999. “The COOH-Terminal Domain of Myo2p, a Yeast Myosin V, has a Direct Role in Secretory Vesicle Targeting.” *The Journal of Cell Biology* 147 (4): 791–808.

Dear Ravi,

Thank you for contacting me with a provisional revision plan for your manuscript. I apologise for the delay in responding to you due to the high number of submissions that we are currently receiving.

I have now looked into your point-by-point response, and I am glad to see that you are willing to engage in a major revision. However, since the outcome of the more far-reaching experiments that you propose is difficult to predict, I am afraid that I cannot explicitly offer to invite a revised manuscript. However, should the proposed experiments be successful, I would be happy to reconsider the manuscript, while treating it as a new submission. In this case, depending on the added findings, I would send it back to the original reviewers. Due to the substantial amount of new results that would have to be added, I would allow them to make new comments on the data, which might then have to be further addressed.

I appreciate that you contacted us for further discussion of your work, and I hope that the proposed approach sounds reasonable to you.

With kind regards,

William

William Teale, PhD
Editor
The EMBO Journal
w.teale@embojournal.org

** As a service to authors, EMBO Press provides authors with the possibility to transfer a manuscript that one journal cannot offer to publish to another EMBO publication or the open access journal Life Science Alliance launched in partnership between EMBO Press, Rockefeller University Press and Cold Spring Harbor Laboratory Press. The full manuscript and if applicable, reviewers' reports, are automatically sent to the receiving journal to allow for fast handling and a prompt decision on your manuscript. For more details of this service, and to transfer your manuscript please click on Link Not Available. **

Referee #1

We sincerely thank Reviewer 1 for the thoughtful and constructive comments provided on our manuscript. We have carefully addressed all the points raised, and the corresponding changes in the revised manuscript are indicated by **line numbers** for clarity. In this document, the reviewer's comments are presented in **blue**, and our point-by-point rebuttals are provided in **black**.

The current study by Nambiar et al. builds on recent findings from the Manjithaya group, which investigated the role of general trafficking factors in autophagy. In this work, the authors focused on an unconventional class V myosin, MYO5A, and its role in regulating the fusion between autophagosomes and lysosomes. Using three biological models—yeast, mammalian cells, and *C. elegans*. They provide evidence supporting MYO5A's involvement in autophagy. The authors analysed different domains of MYO5A and found that the CC-GTD domain (amino acids 923-1829) associates with LC3, a protein that marks mature autophagosomes. This interaction increases the number of stationary autophagosomes when directional movement is inhibited, suggesting a role in their consumption.

1. While the proposed model for MYO5A's role in regulating autophagosome movement is promising, many results are descriptive, and the data do not fully support the suggested mechanisms. For instance, the study emphasizes the association between MYO5A and autophagosomes via LC3 interactions. The authors identified six predicted LIRs on the CC-GTD domain of MYO5A using the iLIR server. Still, they did not validate these interactions with LC3 or their role in inhibiting autophagosome movement.
- We thank the reviewer for this valuable suggestion, which has added mechanistic depth to our study. To address it, we examined the functional relevance of predicted LIR motifs in MYO5A. Sequence analysis (iLIR + ANCHOR) identified six conserved motifs, four within the coiled-coil domain and two in the globular tail domain. From these, two motifs in the CC-GTD region (P: PAYRVL; Q: QAYIGL) were selected for further study, as this region was previously shown to be critical for autophagosome association.
 - Using single (P, Q) and double (DM) alanine mutants, we found that all LIR mutants showed reduced colocalization with LC3 but not with LAMP1. Full-length MYO5A and CC-GTD rescued LC3-LAMP1 colocalization in KO cells, while the Q mutant displayed a strong defect, and the double mutant failed completely. Immunoprecipitation further confirmed reduced LC3 binding by Q and DM mutants. These results (**line no. 394-435**), now incorporated as **Figure 6**, demonstrate that MYO5A directly engages LC3 through its LIR motifs to mediate autophagosome-lysosome fusion.
 - In addition, live-cell tracking revealed that while CC-GTD and single mutants could restore excessive directional switching of autophagosomes in KO cells, the double mutant could not. This analysis (**line no. 463-472**), now presented in **Figure 7d-e**, shows that LIR-mediated LC3 engagement is also required for stabilizing autophagosome dynamics. Collectively, these findings provide mechanistic insight into how MYO5A couples LC3 recognition with motor activity to ensure fusion-competent autophagosome behaviour.

Figure 6. LIR motifs in MYO5A mediate its interaction with autophagosomes and are required for autophagosome–lysosome fusion

(a) Schematic representation of the MYO5A protein. The black boxes indicate the positions of LIR motifs mapped to the coiled-coil and globular tail domains. The positional information of the LIR motifs was analyzed using the iLIR web tool and is depicted below. On the right, the conservation of the LIR motifs across organisms is shown. (b) *MYO5A* KO cells were transfected with GFP-MYO5A, mEmerald-CC-GTD, or mEmerald-tagged P, Q, and DM LIR mutants in the CC-GTD domain of MYO5A, and co-transfected with mCh-LC3. Zoomed insets highlight co-localization events between mCh-LC3 and MYO5A constructs, as well as between mCh-LC3 and LAMP1. Scale bar, 1 μ m. (c–e) Quantification of co-localization events between LC3 and MYO5A (c), LAMP1 and MYO5A (d), and LC3 and LAMP1 (e) in *MYO5A* KO cells transfected with different LIR mutant constructs of MYO5A. Analysis was performed using the “Colocalization” plug-in of ImageJ. Data are plotted as mean \pm SEM from three independent experiments (N = 3, n = 60 cells). Statistical significance was assessed by one-way ANOVA with Kruskal–Wallis post-hoc test (**p < 0.01, ***p < 0.001, ns, non-significant). (f) *MYO5A* KO cells were transfected with mEmerald-CC-GTD or CC-GTD domains harbouring different LIR mutants (as indicated). Immunoprecipitation was performed using anti-GFP antibody, and lysates were probed for LC3 (N = 4). (g) Quantification of LC3 levels immunoprecipitated with GFP-tagged CC-GTD or LIR mutant constructs of MYO5A. Data are plotted as mean \pm SEM from three independent experiments (N = 4). Statistical significance was assessed by one-way ANOVA with Tukey post-hoc test.

Figure 7

Figure 7. Mean directional switches of autophagosomes are altered in the absence of MYO5A

(d) Time-lapse images of *MYO5A* KO cells co-transfected with mCh-LC3 and different LIR mutant constructs of MYO5A (as indicated). Scale bar, 1 μ m. (e) Quantification of mean directional changes of LC3-positive autophagosomes in *MYO5A* KO cells transfected with different LIR mutant constructs, analyzed using the “TrackMate” plug-in. Data are plotted as mean \pm SEM from three independent experiments (N = 3, n = 40 cells). Statistical significance was assessed by one-way ANOVA with Kruskal–Wallis post-hoc test (***p < 0.001, **p < 0.01, ns, non-significant).

2. Additionally, the authors suggest that MYO5A interacts with lysosomes through LAMP1, which could be crucial for autophagosome-lysosome fusion. However, this potential interaction was not explored in the study.

We thank the reviewer for raising this important point. We agree that a potential interaction between MYO5A and LAMP1 could, in principle, be relevant for regulating autophagosome-lysosome fusion. To address this, our study systematically assessed lysosomal integrity and dynamics to determine whether the fusion defect in MYO5A-depleted cells arises from lysosomal dysfunction.

1. **Lysosomal integrity and function remain intact in the absence of MYO5A (Fig. 3; lines 227–248).**

We examined lysosomal morphology, acidification, positioning, and proteolytic competence in WT and *MYO5A* KO cells. Both imaging and immunoblotting approaches showed that lysosomal parameters (LAMP1, RAB7, Cathepsin D, LysoTracker staining, radial distribution, and EGFR degradation assays) were unaffected by *MYO5A* loss. These findings argue against a role for *MYO5A* in lysosomal maintenance or function.

Figure 3. Lysosomal acidification, positioning, and morphology remain unperturbed upon *MYO5A* knockout

(a) WT and *MYO5A* KO HeLa cells were either untreated or subjected to starvation (EBSS, 2 h), followed by LysoTracker Deep Red (LTDR) staining or antibody labelling against LAMP1 and RAB7. Scale bar, 10 μ m. Quantification of mean intensity per cell for (b) RAB7, (c) LTDR-positive, and (d) LAMP1 puncta was performed using the *Measure* plug-in of FIJI ImageJ, and plotted as mean \pm SEM from three independent experiments (N = 3, n = 100 cells). Statistical significance was assessed by one-way ANOVA with Bonferroni post-hoc analysis (****p < 0.0001, **p < 0.01). (e) Radial distribution analysis of LAMP1 puncta from the nucleus was quantified using the “Radial Plot” plug-in of FIJI ImageJ, plotted as mean \pm SEM from three independent experiments (N = 3, n = 70 cells). No significant differences were observed between groups (one-way ANOVA with Kruskal–Wallis’s test). (f) Lysates from WT and *MYO5A* KO HeLa cells treated with EBSS (2 h) and/or BafA1 (100 nM, 2 h) were immunoblotted with antibodies against Cathepsin D (CTSD), RAB7, LAMP1, and *MYO5A*. Relative protein levels of (g) CTSD, (h) RAB7, and (i) LAMP1 were quantified from three independent experiments and plotted as mean \pm SEM. Statistical significance was assessed by one-way ANOVA with Bonferroni post-hoc analysis relative to WT (GM) control. (j) WT HeLa cells transfected with either *Scr* or *MYO5A* siRNA were serum-starved for 3 h and subsequently left untreated or pre-treated with PBP before stimulation with EGF (100 ng/ml) for the indicated time points. Lysates were immunoblotted for EGFR. (k) Quantification of EGFR levels from three independent experiments plotted as mean \pm SEM. Statistical significance was determined using one-way ANOVA with Bonferroni post-hoc analysis.

2. ***C. elegans* analysis confirms no gross lysosomal dysfunction (Fig. 8j; lines 511–514).**

To rule out lysosomal abnormalities in vivo, we examined endo-lysosomal compartments in *C. elegans* PLM neurons using CTNS-1::mCh as a lysosomal marker. The number, distribution, and soma-enrichment of lysosomes were indistinguishable between *hum-2(ok596)* mutants and WT animals, further arguing against lysosomal biogenesis or positioning defects.

Figure 8

8j. CTNS-1::mCh in the cell body and proximal process of PLM neurons in WT and *hum-2(ok596)* L4 worms. Scale bar, 10 μ m. quantification of CTNS-1::mCh distribution length from the cell body using the “Measure” plug-in of ImageJ, plotted as mean \pm SEM from three independent experiments (N = 3, n = 30 worms). Statistical significance was assessed by Student’s t-test (ns = non-significant).

3. MYO5A–autophagosome interaction, but not MYO5A–lysosome interaction, is critical (Fig. 6b, 6c; lines 409-413).

Our LIR mutant analyses demonstrated that disruption of the LIR motifs selectively reduced MYO5A–LC3 association, while colocalization with LAMP1 remained unchanged. This strongly suggests that MYO5A primarily couples to autophagosomes rather than lysosomes to mediate fusion.

6d. Quantification of co-localization events LAMP1 and MYO5A in *MYO5A* KO cells transfected with different LIR mutant constructs of MYO5A. Analysis was performed using the “Colocalization” plug-in of ImageJ. Data are plotted as mean \pm SEM from three independent experiments (N = 3, n = 60 cells). Statistical significance was assessed by one-way ANOVA with Kruskal–Wallis post-hoc test (**p < 0.01, ***p < 0.001, ns, non-significant).

4. Live-cell imaging confirms that MYO5A loss alters autophagosome dynamics, not lysosomal motility (Fig. 7a–c; Supp. Fig. 5; lines 443–462).

- Tracking assays revealed no significant changes in lysosome speed, displacement, or trajectory between WT and *MYO5A* KO cells. By contrast, autophagosomes in *MYO5A* KO cells displayed increased directional switching, a defect rescued by full-length MYO5A or the CC–GTD domain. This directional instability likely reduces productive encounters with lysosomes, thereby impairing fusion.
- Together, these complementary datasets provide compelling evidence that MYO5A does not regulate lysosomal function or motility, and that a direct MYO5A–LAMP1 interaction is unlikely to be required for fusion. Instead, our findings position LIR-mediated engagement of MYO5A with autophagosomes as the critical determinant of their fusion competence. To avoid ambiguity, we have revised the Results and Discussion to emphasize this conclusion.

Figure 7. Mean directional switches of autophagosomes are altered in the absence of MYO5A
(a) Time-lapse images of WT and *MYO5A* KO cells co-transfected with mCh-LC3 and different MYO5A constructs (as indicated), and stained with LysoTracker Deep Red dye. Scale bar, 1 μ m.
(b–c) Quantification of mean directional changes of LC3-positive autophagosomes **(b)** and lysosomes **(c)** using the “TrackMate” plug-in of ImageJ. Data were averaged for each cell and plotted as mean \pm SEM from three independent experiments (N = 3, n = 40 cells). Statistical significance was assessed by one-way ANOVA with Kruskal–Wallis post-hoc test (** $p < 0.001$, * $p < 0.1$, ns, non-significant).

Supplementary figure 5. Quantification of motion properties of autophagosomes and lysosomes

Time-lapse images were acquired from WT and *MYO5A* KO cells co-transfected with mCh-LC3 and different MYO5A constructs (as indicated in Fig.7) and stained with LysoTracker Deep Red dye. Graphs represent **(a)** mean speed of LC3 and lysosomes, **(b)** total distance traveled by LC3 and lysosomes, and **(c)** total displacement of LC3 and lysosomes, quantified using the “TrackMate” plug-in of ImageJ and averaged for each cell. Data are plotted as mean \pm SEM from three independent experiments (N = 3, n = 40 cells) across different cohorts. Statistical significance was assessed by one-way ANOVA with Kruskal–Wallis post-hoc test (** $p < 0.001$, * $p < 0.1$, ns, non-significant).

- Overall, the hypothesis regarding MYO5A's role in forming competent SNARE and HOPS fusion complexes requires significant experimental validation, lacking in this research.

We very much appreciate the reviewer's thoughtful comment regarding the need for mechanistic validation of MYO5A's role in facilitating SNARE and HOPS complex assembly. In response, we have performed additional experiments to address this point.

1. **Defective SNARE assembly in the absence of MYO5A (Fig. 4g and Fig. 4h; lines 295-299).**

We specifically asked whether the fusion block observed in MYO5A KO cells could stem from impaired SNARE assembly. To this end, we immunoprecipitated Flag-VAMP8 and examined its interactions with its cognate SNARE partners STX17 and SNAP29. Consistent with our earlier findings, MYO5A KO cells exhibited a reduced association of VAMP8 with both STX17 and SNAP29, strongly suggesting a defect in functional SNARE complex assembly (Fig. 4g and Fig. 4h).

Figure 4

Figure 4g. WT and MYO5A KO cells were transfected with Flag-VAMP8 and Flag-SNAP29 and either left untreated or subjected to starvation (EBSS, 2 h). Lysates collected 48 h post-transfection were subjected to immunoprecipitation with anti-Flag antibody, followed by immunoblotting with the indicated antibodies. **(h)** Quantification of the proportion of Flag-SNAP29 and STX17 co-precipitated with Flag-VAMP8 from WT and MYO5A KO cells under basal or starvation conditions. Data are plotted as mean \pm SEM from three independent experiments. Statistical significance was determined by one-way ANOVA with Tukey post-hoc analysis (* $p < 0.1$).

2. **Requirement of full-length MYO5A for engaging fusion machinery (Supp. Fig. 4e-g; lines 379-390).**

- To further test whether specific MYO5A domains contribute to fusion complex engagement, we reintroduced either full-length MYO5A or the CC-GTD fragment into MYO5A KO cells and examined colocalization of LC3 with the late fusion regulators STX17 and VPS33A. Re-expression of full-length MYO5A successfully restored colocalization of LC3 with both STX17 and VPS33A, whereas the CC-GTD fragment failed to rescue these interactions (Supp. Fig. 4e, Supp. Fig. 4f, and Supp. Fig. 4g). This demonstrates that although the CC-GTD domain is sufficient to target MYO5A to autophagic vesicles, the full-length protein is essential for coordinating the recruitment of the fusion machinery, thereby ensuring productive autophagosome-lysosome fusion.
- Together, these findings provide direct mechanistic insight, validating the hypothesis that MYO5A is required for competent SNARE and HOPS complex formation. We have revised the Results and Discussion to highlight these findings more explicitly.

Supplementary fig. 4e. WT and *MYO5A* KO cells (as indicated) were either left untransfected or transfected with GFP-MYO5A or mEmerald-CC-GTD plasmid constructs and co-transfected with HA-VPS33A or Flag-STX17. Zoomed insets highlight co-localization events between VPS33A or STX17 (top) and LC3, as well as expression of MYO5A constructs in the cell (bottom). Scale bar, 1 μ m. **f, g.** Quantification of co-localization events between VPS33A (**f**) or STX17 (**g**) and LC3 in WT and *MYO5A* KO cells using the Colocalization plug-in of ImageJ. Data are plotted as mean \pm SEM from three independent experiments ($N = 3$, $n = 60$ cells). Statistical significance was assessed by one-way ANOVA with Kruskal-Wallis post-hoc test (**** $p < 0.0001$, *** $p < 0.001$, ns, non-significant).

- The authors should complement their western blot results with immunofluorescence (IF) experiments, as mentioned in Fig. 1a. This aspect of the study needs further exploration to clarify the nature of the intermediates that accumulate in the absence of Myo2.

We thank the reviewer for this valuable suggestion. To complement our western blot analysis and clarify the nature of the intermediates that accumulate in the absence of Myo2, we have now performed fluorescence microscopy experiments to visualize both general and selective autophagy.

- General autophagy (Fig. 1h–o; lines 148-156).** We monitored GFP-Atg8 localization in WT, *myo2 ts* mutant alleles, and *atg1A* cells. Vacuoles were stained with CMAC blue dye, and the percentage of cells displaying punctate versus vacuolar GFP signal was quantified. In WT cells, starvation for 4 h at PT led to robust accumulation of GFP signal in the vacuole, consistent with efficient autophagic flux. By contrast, *Myo2 ts* mutants displayed reduced vacuolar GFP at NPT, accompanied by increased punctate cytoplasmic GFP-Atg8 structures, indicative of stalled autophagosomes. As expected, *atg1A* cells exhibited persistent punctate GFP-Atg8 under both PT and NPT, reflecting their inability to form functional autophagosomes.

Figure 1h-m. Representative fluorescence micrographs of GFP-Atg8 localization. Vacuoles were stained with CMAC blue. **(n-o)** Quantification of cells showing punctate versus vacuolar GFP signal. WT cells showed strong vacuolar GFP accumulation after starvation at PT, while *myo2* mutants at NPT retained punctate cytoplasmic GFP-Atg8. *atg1Δ* cells exhibited persistent puncta under both conditions. Data represent mean \pm SEM from three independent experiments ($n \geq 100$ cells per condition). Statistical significance was determined using one-way ANOVA with Bonferroni post-hoc test (** $p < 0.01$, *** $p < 0.001$).

2. **Selective autophagy – pexophagy (Supp. Fig. 1g-j; lines 164-170).** To determine whether this requirement extends to selective pathways, we examined pexophagy using GFP as a peroxisomal marker. At the single-cell level, vacuoles were again stained with CMAC blue dye, and the percentage of cells showing vacuolar GFP was quantified. In WT cells, clear vacuolar GFP accumulation was observed after 6 h of starvation at both PT and NPT, reflecting efficient delivery of peroxisomes to the vacuole. Strikingly, the *myo2-13* mutant showed a pronounced reduction in vacuolar GFP accumulation at NPT, indicating defective pexophagy. As expected, *atg1Δ* cells displayed no vacuolar GFP signal under either condition.

Together, these fluorescence microscopy results complement our immunoblotting data and reinforce the conclusion that Myo2 is required not only for bulk autophagy but also for selective autophagy (pexophagy).

Supplementary figure. (g-i) Representative fluorescence micrographs of Pot1-GFP localization in WT (*MYO2*), *atg1Δ*, and *myo2-13* strains after oleic acid induction, followed by 6 hrs. starvation. Vacuoles were stained with CMAC blue. Scale bar: 5 μ m. **(j)** Quantification of cells displaying vacuolar Pot1-GFP signal. Data represent mean \pm SEM from three independent experiments ($n \geq 100$ cells per condition). Statistical significance was determined using one-way ANOVA with Bonferroni post-hoc test (***) $p < 0.001$.

5. Figure 2: In line 147, it states: "MYO5AKO cells, upon PBP treatment, were comparatively larger in size..."-however, the knockout cells were not treated with PBP, at least as presented.

We thank the reviewer for pointing out this error. We apologize for the confusion in our initial text. The sentence has now been corrected to accurately reflect the experimental design. The revised version reads:

"In WT cells, accumulation of autolysosomes was observed upon starvation (EBSS), consistent with efficient flux (Fig. 2a, Fig. 2b, and Fig. 2c). By contrast, MYO5A KO cells and cells treated with PBP displayed a significant increase in autophagosomes together with a reduction in autolysosomes, indicative of impaired flux (Fig. 2a, Fig. 2b, and Fig. 2c)."

This correction appears in the revised manuscript (lines 190–193).

6. Figure 5: The protection assay should be conducted in MYO5A knockout (KO) cells instead of relying on siRNA.

We thank the reviewer for this constructive suggestion. The protease protection assay was originally performed in *MYO5A* knockdown (KD) cells, as these experiments were conducted before our *MYO5A* knockout (KO) line became available. Importantly, the phenotypes observed in KO cells, including a

block in autophagic flux and related defects, closely mirror those seen in the KD cells. Since our siRNA-mediated knockdown achieves ~85% depletion of MYO5A, we reason that the protease protection assay results in KO cells would be highly consistent with those obtained in KD cells.

7. Figure 7b: The choice of colours is misleading; cells expressing GFP-MYO5A (which should appear green) are represented using mCh-LC3. Additionally, the localization of all three proteins needs to be determined.

- We thank the reviewer for this valuable observation. For clarity, we used pseudo-colours (green and magenta) as they are colour-blind friendly and allow colocalization events to be readily visualized as white. These choices have now been explicitly clarified in the figure legends, and Materials & Methods sections.
- We also appreciate the reviewer's suggestion regarding triple co-localization analysis. In response, we have now performed these experiments to simultaneously examine the localization of GFP-MYO5A, mCh-LC3, and LAMP1. The new data have been included, and they support our earlier conclusion that MYO5A associates with autophagosomes and contributes to their engagement with lysosomes.

Figure: Quantification of co-localization events between LC3, LAMP1 and MYO5A in WT HeLa cells using the Colocalization plug-in of ImageJ. Data are plotted as mean \pm SEM from three independent experiments (N = 3, n = 60 cells).

We hope that the revisions, additional analyses, and clarifications provided in the revised manuscript satisfactorily address Reviewer 1's concerns. We are grateful for the insightful feedback, which has significantly strengthened the quality and clarity of our study.

Referee #2:

We sincerely thank Reviewer 2 for the highly thoughtful, detailed, and constructive comments on our manuscript. Your suggestions have been invaluable in helping us refine both the clarity and depth of our study. We have carefully considered and addressed each point raised, and the corresponding changes in the revised manuscript are indicated by **line numbers** for ease of reference. In this document, the reviewer's comments are presented in **blue**, and our point-by-point rebuttals are provided in **black**.

Roles of unconventional myosins in autophagy are less characterized. Here, Nambiar *et al.*, reveal that class V myosin, MYO5A is a critical regulator of autophagy by utilizing several models, including yeast, worms, and mammalian cells. They demonstrate that inhibition of MYO5A impairs autophagosome-lysosome fusion. Mechanistically, MYO5A is required for the recruitment of several autophagy components which is essential for the fusion on autophagosomes. They showed that MYO5A interacts with autophagosomes through its coiled-coiled and globular tail domains. Live imaging analysis indicates that MYO5 is critical to increase stationary autophagosomes. Although the data shown could be potentially interesting, the manuscript is very descriptive and lack clear mechanistic insights. The specific comments are listed below.

1. It is unclear which functions of MYO5 is critical to regulate autophagosome-lysosome fusion. In Figure 5e and Figure 8, it is difficult to understand the mechanism in which the regulation of directional switch of autophagosome by MYO5A contributes to the interaction between LC3 and VPS33A or STX17. In addition, the authors need to elucidate whether the re-introduction of FL and CC-GTD constructs in MYO5A KO cells also improves the decreased interactions of STX17 and VPS33A with LC3
- To address the reviewer's concern regarding which functions of MYO5A are critical for autophagosome-lysosome fusion, we performed domain reintroduction experiments in *MYO5A* KO cells. We reintroduced either full-length MYO5A or the CC-GTD fragment and examined colocalization of LC3 with the late fusion regulators STX17 and VPS33A. Strikingly, re-expression of full-length MYO5A successfully restored colocalization of LC3 with both STX17 and VPS33A, whereas the CC-GTD fragment failed to rescue these interactions (**Supp. Fig. 4e-g; lines 379-390**). These findings demonstrate that although the CC-GTD domain is sufficient to target MYO5A to autophagic vesicles, the full-length protein is indispensable for coordinating recruitment of the fusion machinery, thereby ensuring productive autophagosome-lysosome fusion.

Supplementary fig. 4e. WT and *MYO5A* KO cells (as indicated) were either left untransfected or transfected with GFP-MYO5A or mEmerald-CC-GTD plasmid constructs and co-transfected with HA-VPS33A or Flag-STX17. Zoomed insets highlight co-localization events between VPS33A or STX17 (top) and LC3, as well as expression of MYO5A constructs in the cell (bottom). Scale bar, 1 μm . f, g. Quantification of co-localization events between VPS33A (f) or STX17 (g) and LC3 in WT and *MYO5A* KO cells using the Colocalization plug-in of ImageJ. Data are plotted as mean \pm SEM from three independent experiments (N = 3, n = 60 cells). Statistical significance was assessed by one-way ANOVA with Kruskal–Wallis post-hoc test (****p < 0.0001, ***p < 0.001, ns, non-significant).

- We agree with the reviewer that the precise mechanism by which regulation of directional switching by MYO5A influences SNARE and HOPS engagement remains to be elucidated. While we have not directly tested this connection in the current study, our results clearly establish that MYO5A is necessary for proper assembly of the fusion machinery. We have revised the Results and Discussion sections to explicitly emphasize this point (the Discussion section; **lines 567-573**, now reads as, “*A limitation of our study is that we cannot yet resolve whether the stationary state of autophagosomes imposed by MYO5A is itself required for the loading of fusion machinery, or whether pausing and machinery recruitment occur as independent steps. Although our rescue experiments suggest that reintroduction of full-length MYO5A restores VPS33A and STX17 localization to autophagosomes, the causal relationship between pausing and fusion machinery assembly remains unresolved. Dissecting this temporal sequence will require mechanistically higher-resolution approaches, such as acute perturbation of MYO5A-actin interactions in live imaging systems*”). We believe this important mechanistic question warrants dedicated investigation in the future, and our present findings lay the groundwork for such studies.
2. It is important to show if the mutations in LIR of MYO5A affect the autophagosome-lysosome fusion and the pool of stationary autophagosomes.
- We thank the reviewer for this valuable suggestion, which has added mechanistic depth to our study. To address it, we examined the functional relevance of predicted LIR motifs in MYO5A. Sequence analysis (iLIR + ANCHOR) identified six conserved motifs, four within the coiled-coil domain and two in the globular tail domain. From these, two motifs in the CC-GTD region (P: PAYRVL; Q: QAYIGL) were selected for further study, as this region was previously shown to be critical for autophagosome association.
 - Using single (P, Q) and double (DM) alanine mutants, we found that all LIR mutants showed reduced colocalization with LC3 but not with LAMP1. Full-length MYO5A and CC-GTD rescued LC3–LAMP1 colocalization in KO cells, while the Q mutant displayed a strong defect, and the double mutant failed completely. Immunoprecipitation further confirmed reduced LC3 binding by Q and DM mutants. These results (**line no. 394-435**), now incorporated as **Figure 6**, demonstrate that MYO5A directly engages LC3 through its LIR motifs to mediate autophagosome–lysosome fusion.
 - In addition, live-cell tracking revealed that while CC-GTD and single mutants could restore excessive directional switching of autophagosomes in KO cells, the double mutant could not. This analysis (**line no. 463-472**), now presented in **Figure 7d–e**, shows that LIR-mediated LC3 engagement is also required for stabilizing autophagosome dynamics. Collectively, these findings provide mechanistic insight into how MYO5A couples LC3 recognition with motor activity to ensure fusion-competent autophagosome behaviour.

Figure 6. LIR motifs in MYO5A mediate its interaction with autophagosomes and are required for autophagosome–lysosome fusion

(a) Schematic representation of the MYO5A protein. The black boxes indicate the positions of LIR motifs mapped to the coiled-coil and globular tail domains. The positional information of the LIR motifs was analyzed using the iLIR web tool and is depicted below. On the right, the conservation of the LIR motifs across organisms is shown. (b) *MYO5A* KO cells were transfected with GFP-MYO5A, mEmerald-CC-GTD, or mEmerald-tagged P, Q, and DM LIR mutants in the CC-GTD domain of MYO5A, and co-transfected with mCh-LC3. Zoomed insets highlight co-localization events between mCh-LC3 and MYO5A constructs, as well as between mCh-LC3 and LAMP1. Scale bar, 1 μ m. (c–e) Quantification of co-localization events between LC3 and MYO5A (c), LAMP1 and MYO5A (d), and LC3 and LAMP1 (e) in *MYO5A* KO cells transfected with different LIR mutant constructs of MYO5A. Analysis was performed using the “Colocalization” plug-in of ImageJ. Data are plotted as mean \pm SEM from three independent experiments (N = 3, n = 60 cells). Statistical significance was assessed by one-way ANOVA with Kruskal–Wallis post-hoc test (**p < 0.01, ***p < 0.001, ns, non-significant). (f) *MYO5A* KO cells were transfected with mEmerald-CC-GTD or CC-GTD domains harbouring different LIR mutants (as indicated). Immunoprecipitation was performed using anti-GFP antibody, and lysates were probed for LC3 (N = 4). (g) Quantification of LC3 levels immunoprecipitated with GFP-tagged CC-GTD or LIR mutant constructs of MYO5A. Data are plotted as mean \pm SEM from three independent experiments (N = 4). Statistical significance was assessed by one-way ANOVA with Tukey post-hoc test.

Figure 7

Figure 7. Mean directional switches of autophagosomes are altered in the absence of MYO5A

(d) Time-lapse images of *MYO5A* KO cells co-transfected with mCh-LC3 and different LIR mutant constructs of MYO5A (as indicated). Scale bar, 1 μ m. (e) Quantification of mean directional changes of LC3-positive autophagosomes in *MYO5A* KO cells transfected with different LIR mutant constructs, analyzed using the “TrackMate” plug-in. Data are plotted as mean \pm SEM from three independent experiments (N = 3, n = 40 cells). Statistical significance was assessed by one-way ANOVA with Kruskal–Wallis post-hoc test (***p < 0.001, **p < 0.01, ns, non-significant).

3. In the introduction part, the authors mention that MYO5A mutation causes the defect in pexophagy. Is MYO5A also required for the autophagosome-lysosome fusion during several selective autophagy such as mitophagy, lipophagy and ERphagy?

- We thank the reviewer for this insightful comment. Using a Pot1–GFP pexophagy assay, we found that conditional Myo2 *ts* mutants showed markedly reduced free GFP release and vacuolar accumulation compared to WT, confirming a requirement for Myo2 in selective autophagy (Supp. Fig. 1a–j; lines 158–171). These findings, together with GFP-Atg8 assays, establish that Myo2 is required for both bulk and selective autophagy in yeast. In mammalian cells, loss of MYO5A similarly impaired autophagosome–lysosome fusion. Since MYO5A/Myo2 disruption blocks fusion itself, we expect this defect to extend to multiple selective autophagy pathways, including mitophagy, lipophagy, and ER-phagy, although these were not directly assayed in the present study.

Supplementary figure 1. Myo2 is required for pexophagy in *Saccharomyces cerevisiae*

(a) Schematic of the Pot1-GFP processing assay for pexophagy. Cells expressing Pot1-GFP were grown in oleate medium to promote peroxisome proliferation and then starved in SD-N to induce autophagy-dependent degradation of peroxisomes. Free GFP accumulation in the vacuole was monitored by immunoblotting. **(b)** Pot1-GFP processing in *atg1Δ* cells as an autophagy-deficient control, where free GFP release was absent. **(c-f)** Pot1-GFP processing in WT (*MYO2*) and *myo2 ts* alleles (*myo2-12*, *myo2-13*, *myo2-16*) under PT and NPT conditions. Lysates were collected at indicated time points and analysed by immunoblotting. Graphs show free GFP/Pot1-GFP ratios (arbitrary units), mean of three independent experiments (error bars: SEM). Statistical significance was determined using one-way ANOVA with Bonferroni post-hoc test (**** $p < 0.0001$, ** $p < 0.01$). **(g-i)** Representative fluorescence micrographs of Pot1-GFP localization in WT (*MYO2*), *atg1Δ*, and *myo2-13* strains after oleic acid induction, followed by 6 hrs. starvation. Vacuoles were stained with CMAC blue. Scale bar: 5 μ m. **(j)** Quantification of cells displaying vacuolar Pot1-GFP signal. Data represent mean \pm SEM from three independent experiments ($n \geq 100$ cells per condition). Statistical significance was determined using one-way ANOVA with Bonferroni post-hoc test (**** $p < 0.001$).

4. Related to Figure 4i, is the increased Igg-1 puncta in the proximal process in *hum-2* suppressed by expression of exogenous HUM-2 in *C. elegans*? Similarly, the other phenotypes such as no. of embryos laid and percentage of worm survival in *hum-2* in Supp. Figure 3h and 3i are also rescued by expression of exogenous HUM-2?

- We thank the reviewer for this insightful comment. We did not perform a rescue experiment on autophagic defects or life-history traits through exogenous overexpression of hum-2, as we faced technical limitations in achieving stable and robust expression of HUM-2 in the *hum-2(ok596)* mutant background. However, to address this point, we examined the heterozygous *hum-2(ok596)* mutant, in which one copy of the mutant allele was replaced with the wild-type allele by crossing with wild-type worms.
- In these heterozygotes, the reduced brooding capacity was restored (see figure below). In addition, we analysed the status of lgg-1 puncta in the cell body and proximal process of 7-day adult worms, which were also rescued in the heterozygous background (see figure below).
- These findings support the idea that reintroduction of functional HUM-2 can reverse the observed defects. While these results provide indirect evidence for the specificity of the phenotype, a direct rescue through exogenous HUM-2 expression remains an important future direction.

Figure: Brood size indicates the ability of *C. elegans* to produce offspring. The graph represents the number of embryos laid by an adult WT (N2) and *hum-2(ok596)/+* heterozygous mutant worms and is plotted and represented as mean \pm SEM between the genotypes (N=3, n=30 worms). Statistical significance was assessed by unpaired Student's t-test (ns, non-significant).

Figure: Quantification of mNG::LGG-1 puncta in the cell body and proximal process of PLM neurons in 7-day adult worms. Graphs are plotted as mean \pm SEM from three independent experiments (N = 3, n = 40 worms). Statistical significance was assessed by unpaired Student's t-test (ns, non-significant).

5. The author should carefully make the Synopsis. They show the VAMP8 on the lysosome which interacts with SNAP29 in Synopsis but there is no data about the VAMP8 in this paper.
- We sincerely apologize for the confusion caused by our Synopsis. The reviewer is correct that the Synopsis gave the impression that VAMP8 findings were not part of the manuscript.
 - To clarify, we did in fact examine VAMP8 as part of our study. Specifically, we observed that the presence of the R-SNARE VAMP8 on lysosomes was unaltered, as demonstrated by its colocalization with LAMP1 in both WT and *MYO5A* KO cells (**Fig. 4c and Fig. 4d; lines 275-281**).

Figure 4

Figure 4c. WT and *MYO5A* KO cells were transfected with Flag-VAMP8 and stained for LAMP1. Scale bar, 10 μ m. Insets show co-localization events between Flag-VAMP8 and LAMP1. Scale bar, 1 μ m. **(d)** Quantification of co-localization between LAMP1 and Flag-VAMP8 in WT and *MYO5A* KO cells. Data represent mean \pm SEM from three independent experiments (N = 3, n = 60 cells). Statistical significance was assessed by unpaired Student's t-test (ns, non-significant).

- Furthermore, to test whether the reduced recruitment of fusion machinery reflected defective SNARE assembly, we immunoprecipitated Flag-VAMP8 and assessed its interactions with its cognate partners STX17 and SNAP29. Consistent with our hypothesis, *MYO5A* KO cells displayed significantly reduced association of VAMP8 with both STX17 and SNAP29 (**Fig. 4g and Fig. 4h; lines 296-300**). Together, these results demonstrate that while VAMP8 is present

and correctly localized on lysosomes, MYO5A is essential for productive SNARE assembly. We have revised the Synopsis text to ensure that it accurately reflects the data presented and avoids any misleading interpretation.

Figure 4

Figure 4g. WT and *MYO5A* KO cells were transfected with Flag-VAMP8 and Flag-SNAP29 and either left untreated or subjected to starvation (EBSS, 2 h). Lysates collected 48 h post-transfection were subjected to immunoprecipitation with anti-Flag antibody, followed by immunoblotting with the indicated antibodies. **(h)** Quantification of the proportion of Flag-SNAP29 and STX17 co-precipitated with Flag-VAMP8 from WT and *MYO5A* KO cells under basal or starvation conditions. Data are plotted as mean \pm SEM from three independent experiments. Statistical significance was determined by one-way ANOVA with Tukey post-hoc analysis (* $p < 0.1$).

- The author needs to quantify the efficiency of the entrapped lysosome to autophagosome in Figure 2e, perinuclear clustering lysosome in Figure 3a, and the colocalization of the VPS33A with the autophagosomes in Figure 5d. Figure 3 may be better to move to the Supp. Instead, Supp Figure 3h and 3i would be better to show as the main Figure. The author should indicate the statistical difference in Figure 3i.
- We thank the reviewer for this constructive comment. In response, we have now quantified (i) the efficiency of autophagosome–lysosome engagement (Fig. 2d and Fig. 2e), (ii) the radial distribution of lysosomes (Fig. 3e), and (iii) the colocalization of LC3 with VPS33A (Fig. 4e and Fig. 4f).
 - In WT cells, mCherry-LC3B-labelled autophagosomes displayed robust association with LAMP1-GFP/LysoTracker-labelled lysosomes, whereas this association was significantly reduced in *MYO5A* KO cells (Fig. 2d and Fig. 2e; lines 197–200).

Figure 1d. Time-lapse imaging of WT and *MYO5A* KO HeLa cells co-expressing mCh-LC3B and LAMP1-GFP, and stained with LysoTracker Deep Red. Scale bar: 1 μ m. Arrows indicate autophagosome (mCh-LC3B) - lysosome (LysoTracker^{+ve}) fusion events. **(e)** Quantification of LC3B puncta colocalizing with LysoTracker^{+ve} lysosomes using the “Colocalization” plug-in in ImageJ. Data represent mean \pm SEM from three independent experiments (N = 3, n = 40 cells). Statistical significance was assessed by Student’s t-test (***p < 0.001).

- Radial distribution analysis revealed perinuclear clustering of lysosomes upon starvation and a dispersed pattern under basal conditions, consistent with physiological expectations and confirming that *MYO5A* does not influence lysosomal positioning (**Fig. 3e; lines 234–237**).

Figure 3e. Radial distribution analysis of LAMP1 puncta from the nucleus was quantified using the “Radial Plot” plug-in of FIJI ImageJ, plotted as mean \pm SEM from three independent experiments (N = 3, n = 70 cells). No significant differences were observed between groups (one-way ANOVA with Kruskal–Wallis’s test).

- Finally, colocalization analysis showed that while autophagosomes in *MYO5A* KO cells were enriched for early-stage markers such as DFCP1, their colocalization with the late fusion regulator VPS33A was markedly reduced (**Fig. 4e and Fig. 4f; lines 280–284**).

Figure 4e. Time-lapse images of WT and *MYO5A* KO HeLa cells transfected with either mCh-LC3B or GFP-LC3B and co-transfected with mCh-DFCP1 or GFP-VPS33A, as indicated. Scale bar, 10 μ m. Insets highlight co-localization events between stage-specific markers and LC3. Scale bar, 1 μ m. (f) Quantification of co-localization between stage-specific markers and LC3B in WT and *MYO5A* KO cells using the Colocalization plug-in in ImageJ. Data represent mean \pm SEM from three independent experiments (N = 3, n = 40 cells). Statistical significance was determined by unpaired Student's t-test (***p < 0.001, **p < 0.01).

- Regarding the suggestion to move Figure 3 to the Supplementary section, we respectfully disagree. We believe that demonstrating lysosomal parameters remain unperturbed in the absence of *MYO5A* is a key finding that strengthens our overall conclusion; namely, that the observed defects arise primarily from impaired autophagosome competence rather than lysosomal dysfunction. For this reason, we consider it important to retain Figure 3 in the main body of the manuscript.
- Finally, in response to the request, we have now added statistical analysis to Figure 3i to indicate significant differences.

7. Are there any reports that other MYOs such as *MYO6* control the directional switch of autophagosome? The author needs to mention about this.

- We thank the reviewer for this comment. We have now expanded both the *Introduction* and *Discussion* to address whether other myosins, contribute to autophagosome regulation.

- Specifically, in the **Introduction (lines 80–88)**, we now highlight that motility alone is not sufficient for fusion, and that vesicle immobilization or spatial anchoring can be equally decisive. For example, Myosin-V has been shown to cluster cargoes at the axon initial segment, and actin patches position dendritic lysosomes to facilitate local degradation (Janssen *et al.*, 2017; Van Bommel *et al.*, 2019).
- These studies raise the possibility that other unconventional myosins, including MYO6, which is known to mediate cargo transport and anchoring in endosomal pathways, may also regulate directional switches of autophagosomes, although this has not yet been directly demonstrated.
- In the **Discussion (lines 545–556)**, we further emphasize that autophagy is not a linear process of vesicle transport but requires timely pauses and stabilization of autophagosomes to ensure productive fusion.
- Our findings support a model where MYO5A functions as a regulatory brake, enforcing pausing at the fusion stage. Similar requirements for vesicle stalling have been reported in other contexts, including HDAC6- and actin-dependent stabilization, INPP5E-mediated actin regulation, and detyrosinated microtubule-based lysosome positioning (Lee *et al.*, 2010; Hasegawa *et al.*, 2016; Mohan *et al.*, 2019). Thus, we place MYO5A within a broader paradigm of cytoskeletal regulation of vesicle pausing and positioning.

8. In Figure 1C, it might be confused to say "Myo2-12 showed significantly higher level of free GFP at PT compared to NPT conditions." You can rewrite correctly for example showing the time.

We thank the reviewer for pointing this out. The sentence has now been revised for clarity and now reads as, "*In WT cells (MYO2), robust free GFP release was observed at both permissive (PT, 25 °C) and non-permissive temperatures (NPT, 36 °C) (Fig. 1c). As expected, autophagy-deficient cells (atg1A) failed to generate free GFP under either condition, validating the specificity of the assay (Fig. 1b). In contrast, Myo2 ts alleles showed a marked reduction in free GFP release at NPT compared to PT, particularly 4 hours after the induction of starvation (Fig. 1d, Fig. 1e, Fig. 1f, Fig. 1g) (lines 144-149)*".

9. There are many discrepancies among the sentence and the figure number. Do Supp. Fig. 1e in line 119, Supp. Fig. 1f in line 121, and Supp. Fig. 1e in line 122 mean Supp. Fig. 1c, 1e, and 1d, respectively? Can you confirm the line 248 and lines 714-720? Does Supp. Figure 4 indicate Supp. Figure 3?

We thank the reviewer for carefully pointing out these discrepancies. All figure numbering inconsistencies have now been thoroughly checked and corrected in the revised manuscript.

10. In line 138, the author mentioned that PBP induces the slowing movement of the MYO5A on the action cytoskeleton. It is more informative if you can show the phenotypes induced by PBP treatment such as the accumulation of the autophagosome may be cancelled by the addition of FL and CC-GTD constructs.

We thank the reviewer for this insightful suggestion. We performed the experiment as recommended; however, we were unable to observe a rescue of the PBP-induced phenotype by expression of either the full-length or the CC-GTD construct of MYO5A. We believe this outcome is informative in itself, as PBP, being a small-molecule inhibitor, exerts a strong and direct effect on the motor activity of MYO5A. Thus, even though the full-length construct contains the motor domain, its function is likely inhibited in the presence of PBP, preventing restoration of autophagic flux. The inability of CC-GTD to rescue further highlights the rationale that the inhibitory effect of PBP is dominant and cannot be bypassed by construct expression.

In summary, we have carefully revised and substantially rewritten the entire manuscript, with particular attention to clarity and conciseness. We have streamlined descriptions to avoid unnecessary detail, while ensuring that all key findings and mechanistic insights are clearly communicated. We believe these revisions have improved both the readability and the overall impact of the manuscript.

References:

- Hasegawa, J. *et al.* (2016) “Autophagosome–lysosome fusion in neurons requires INPP 5E, a protein associated with Joubert syndrome,” *The EMBO Journal*, 35(17), pp. 1853–1867. Available at: <https://doi.org/10.15252/emj.201593148>.
- Janssen, A.F. *et al.* (2017) “Myosin-V induces cargo immobilization and clustering at the axon initial segment,” *Frontiers in Cellular Neuroscience*, 11, p. 260.
- Lee, J.-Y. *et al.* (2010) “HDAC6 controls autophagosome maturation essential for ubiquitin-selective quality-control autophagy,” *The EMBO Journal*, 29(5), pp. 969–980. Available at: <https://doi.org/10.1038/emboj.2009.405>.
- Mohan, N. *et al.* (2019) “Detyrosinated microtubules spatially constrain lysosomes facilitating lysosome–autophagosome fusion,” *Journal of Cell Biology*, 218(2), pp. 632–643. Available at: <https://doi.org/10.1083/jcb.201807124>.
- Van Bommel, B. *et al.* (2019) “F-actin patches associated with glutamatergic synapses control positioning of dendritic lysosomes,” *The EMBO Journal*, 38(15), p. e101183. Available at: <https://doi.org/10.15252/emj.2018101183>.

Referee #3

We sincerely thank Reviewer 3 for the thoughtful and detailed critique of our manuscript. Your comments, particularly regarding the role of the MYO5A motor domain, have been invaluable in strengthening our study. In response, we have carefully revised the manuscript and **rearranged several figures for improved clarity and coherence**, ensuring that the data better align with the proposed model. We have also refined the text to directly address the concerns raised about the motor domain and its contribution to autophagosome–lysosome fusion.

We have carefully addressed all points raised, and the corresponding changes in the revised manuscript are indicated **by line numbers** for clarity. In this document, the reviewer's comments are presented in **blue**, and our point-by-point rebuttals are provided in **black**.

Response removed as the results presented here are unpublished.

Figure for referee with unpublished data and its description has been removed upon request by the authors.

2. Fig 1. They make use of a number of temperature-sensitive mutations in the yeast MYO2 gene. Where are these mutations within the myosin sequence? They compare GFP-Atg8 cleavage at the permissive temperature to the same strain at the restrictive temperature, but this ignores temperature effects on the pathway. It is more typical to compare mutant to WT at each temperature. They incubate the mutants for 4 hrs at the restrictive temperature. This may be too long for these conditionally lethal mutants, resulting in severe effects on cell metabolism and/or integrity. In panel C the mutant exhibits free GFP at 1 hr that then goes down. This is very unusual behaviour. The WT in panel C shows much less cleavage than in panel B, why? In panel F there are two different cleavage products, why?

We thank the reviewer for these detailed comments regarding the design and interpretation of our yeast assays. We address each concern below:

- **Location of mutations within Myo2:** All the temperature-sensitive (*ts*) mutations used in this study map to the tail region of Myo2. These *ts* alleles were kindly provided by Prof. Anthony Bretscher's laboratory (Schott *et al.*, 1999). The specific mutations are as follows:
- ***myo2-16*:** M1212T, L1471S, D1497V
- ***myo2-12*:** H1373R, Q1441L, D1457V, S1512T
- ***myo2-17*:** K1285R, Y1287N, L1436S
- ***myo2-18*:** Y1161F, N1171Y, L1413S, I1453V, I1498M

We have now included these details explicitly in Figure 1 and the main text (**lines 137-139**).

Figure 1a. Schematic of the GFP-Atg8 processing assay for general autophagy. Cells expressing GFP-Atg8 were starved in SD-N medium. Upon autophagosome–vacuole fusion, protease-resistant GFP accumulates in the vacuole and is detected as free GFP by immunoblotting. **(b)** GFP-Atg8 processing in *atg1Δ* cells as an autophagy-deficient control, where free GFP release is absent. **(c–g)** GFP-Atg8 processing in WT (*MYO2*) and *ts myo2* alleles (*myo2-12*, *myo2-16*, *myo2-18*, *myo2-17*) under permissive (PT) and non-permissive (NPT) conditions. Lysates were collected at indicated time points and analysed by immunoblotting. Graphs show free GFP/GFP-Atg8 ratios (arbitrary units), mean of three independent experiments (error bars: SEM). Statistical significance was determined using one-way ANOVA with Bonferroni post-hoc test (* $p < 0.05$, ** $p < 0.01$, *** $p < 0.001$).

- Comparison of mutants vs WT at permissive and non-permissive temperatures:**
 We agree with the reviewer that it is informative to compare mutants to WT at each temperature. In the revised version, we have clarified this in the figure panels and legends, ensuring that WT and mutant comparisons are consistently presented at both permissive (PT, 25 °C) and non-permissive (NPT, 36 °C) conditions.

Figure: Yeast cells expressing GFP-Atg8 were starved in SD-N medium. Starvation induces autophagosomes targeted to the vacuole, and GFP, which is highly protease-resistant, accumulates in the vacuole and is seen as free GFP release upon immunoblotting. The graph depicts the intensity ratio of free GFP against GFP-Atg8 of WT (*MYO2*) and *ts* mutant strains of Myo2 (as indicated) at NPT (non-permissive temperature). The graph represents the mean of three independent experiments (error bars: SEM). Statistical significance was assessed by one-way ANOVA with Bonferroni post-hoc analysis (** $p < 0.01$, *** $p < 0.001$, **** $p < 0.0001$).

- Time points used (4 hrs at restrictive temperature):** We recognize the concern regarding potential effects on cell physiology due to prolonged incubation of conditionally lethal mutants. However, we note that **GFP-Atg8 processing assays in yeast are conventionally performed at 4 hrs after starvation induction** (Klionsky *et al.*, 2021), and our data are consistent with published approaches. To ensure that the cells remained viable under these conditions, we monitored total protein integrity by **Ponceau S staining** (Figure 1, Supp. fig. 1), which demonstrated intact protein content at the analysed time points. While some cell death is inevitable, the majority of cells remained metabolically active, supporting the validity of our readouts.
- Panel C: free GFP appears at 1 hr, then decreases – unusual behaviour:** We agree this is atypical. We believe this reflects a **technical lag in the inactivation of the mutant Myo2 upon temperature shift**. This transient activity could initially permit some free GFP generation, which then diminishes as the mutant becomes fully inactivated.
- Panel C vs Panel B: WT shows less cleavage in C than in B:** Panel B and Panel C correspond to different experimental contexts: Panel B shows GFP-Atg8 processing in WT cells, whereas Panel C presents the same assay in the *myo2-12* *ts* mutant. The apparent difference in cleavage reflects the **distinct dynamics introduced by the *ts* allele**, not a discrepancy in WT behaviour.

- **Panel F: two cleavage products:** The additional lower molecular weight band likely represents **partial proteolytic intermediates of GFP–Atg8**. Importantly, the major band corresponds to the expected free GFP, validating the assay.
3. Fig 2. In panel B the inhibitor results in a large increase in the number of autophagosomes, but the KOs do not, why? In panel D, KO2 results in a much larger increase in the area of LC3 puncta than KO1 or the inhibitor, why? On line 147 they say "Interestingly, we also observed that autophagosomes formed in MYO5A KO cells, upon PBP treatment, were comparatively larger in size". This seems to imply that they are adding the inhibitor to the KO, which makes no sense. In line 176 they mix the order of Lamp1 and Rab7.

We thank the reviewer for these insightful comments regarding Figure 2.

- **Panel B: inhibitor vs KO phenotype differences:** We acknowledge the reviewer’s point that the PBP inhibitor results in a more pronounced accumulation of autophagosomes compared to MYO5A KO cells. This discrepancy likely reflects **the broader impact of PBP on myosin motor activity** beyond MYO5A alone, as PBP is not entirely isoform-specific. In contrast, *MYO5A* KO represents a more targeted perturbation, which may explain the quantitative differences observed.
- **Line 147 phrasing (“autophagosomes formed in MYO5A KO cells, upon PBP treatment, were comparatively larger in size”):** We regret the confusing phrasing. We did not intend to imply that PBP was added on top of MYO5A KO cells. This sentence has been rewritten for clarity in the revised manuscript to avoid such misinterpretation. The revised version reads: *"In WT cells, accumulation of autolysosomes was observed upon starvation (EBSS), consistent with efficient flux (Fig. 2a, Fig. 2b, and Fig. 2c). By contrast, MYO5A KO cells and cells treated with PBP displayed a significant increase in autophagosomes together with a reduction in autolysosomes, indicative of impaired flux (Fig. 2a, Fig. 2b, and Fig. 2c)"*. This correction appears in the revised manuscript (**lines 190–193**).
- **Line 176 (order of LAMP1 and Rab7):** This was a typographical oversight. We have corrected the order consistently throughout the revised manuscript.

4. Fig 3 is mostly negative data in that it shows a variety of parameters that are not affected by MYO5 KO. As such this figure could be moved to the supplement.

We thank the reviewer for this suggestion. However, we respectfully disagree with moving Figure 3 to the Supplementary section. Demonstrating that lysosomal parameters remain unperturbed in the absence of MYO5A is a critical result, as it reinforces our central conclusion that the defects we observe stem primarily from impaired autophagosome competence rather than lysosomal dysfunction. For this reason, we believe it is essential to retain Figure 3 in the main body of the manuscript.

5. Where is the *hum-2(ok596)* mutation within the myosin sequence?

The hypo-morphic mutant strain *hum-2(ok596)* was generated as part of the *C. elegans* Gene Knockout Project at the Oklahoma Medical Research Foundation, a member of the International *C. elegans* Gene Knockout Consortium wherein 2.1kb of the protein has been deleted (<https://cgc.umn.edu/gene/hum-2>) and we also validated the same using PCR. We have revised the manuscript to include this, *“To investigate this, we examined autophagic flux in C.*

elegans touch receptor neurons (TRNs) of *hum-2(ok596)*, a hypo-morphic mutant carrying a 2.1 kb deletion in the *hum-2* locus (lines 487-488)”.

Figure: a. SnapGene map of the *hum-2* gene, showing the location of primers used to differentiate between the wild-type (WT) and deletion mutant *hum-2(ok596)* alleles. The primers are positioned to produce distinct amplicon sizes for each genotype. **b.** Representative agarose gel image of PCR products. WT samples show a 3145 bp band, while *hum-2(ok596)* mutant samples yield a 1045 bp band, confirming successful distinction between the WT and deletion mutant allele.

6. Fig 5. Was the IP performed after detergent solubilization or were they trying to pull down intact organelles? Explain. In panel E, why was STX17 reduced in the KO IP but not Vps33A or SNAP29? What does this mean? Does Myo5 bind to any of these components in vitro using purified proteins or just in whole cell lysates? The model in the Synopsis does not indicate interaction of Myo5 GTD with any specific component, but the text suggests involvement of LIR domains. This is very confusing.

We thank the reviewer for these important comments.

- The immunoprecipitation experiments were performed after detergent solubilization of whole-cell lysates, not intact organelles. The interactions were examined in whole-cell lysates, not using purified proteins.
- While we did observe a reduction in STX17 levels in *MYO5A* KO cells, this decrease was not statistically significant, as also shown in Supplementary Fig. 3. At present, we do not have a clear explanation for this observation.

Supplementary figure 3(a-b). Lysates collected from WT and *MYO5A* KO cells after EBSS (2 hours) and BafA1 (100 nM, 2 hours) treatment were processed and analysed by immunoblotting for different stage-specific markers. Relative levels of ATG2A, WIPI2, ATG5–ATG12, STX17, ATG9A, and FIP200 were quantified from three independent experiments and plotted as mean \pm SEM between different cohorts. Statistical significance was assessed by one-way ANOVA with Bonferroni post-hoc analysis (ns, non-significant).

- We thank the reviewer for this valuable suggestion, which has added mechanistic depth to our study. To address it, we examined the functional relevance of predicted LIR motifs in *MYO5A*. Sequence analysis (iLIR + ANCHOR) identified six conserved motifs, four within the coiled-coil domain and two in the globular tail domain. From these, two motifs in the CC–GTD region (P: PAYRVL; Q: QAYIGL) were selected for further study, as this region was previously shown to be critical for autophagosome association.
- Using single (P, Q) and double (DM) alanine mutants, we found that all LIR mutants showed reduced colocalization with LC3 but not with LAMP1. Full-length *MYO5A* and CC–GTD rescued LC3–LAMP1 colocalization in KO cells, while the Q mutant displayed a strong defect, and the double mutant failed completely. Immunoprecipitation further confirmed reduced LC3 binding by Q and DM mutants. These results (**line no. 394–435**), now incorporated as **Figure 6**, demonstrate that *MYO5A* directly engages LC3 through its LIR motifs to mediate autophagosome–lysosome fusion.
- In addition, live-cell tracking revealed that while CC–GTD and single mutants could restore excessive directional switching of autophagosomes in KO cells, the double mutant could not. This analysis (**line no. 463–472**), now presented in **Figure 7d–e**, shows that LIR-mediated LC3 engagement is also required for stabilizing autophagosome dynamics. Collectively, these findings provide mechanistic insight into how *MYO5A* couples LC3 recognition with motor activity to ensure fusion-competent autophagosome behaviour.

Figure 6

Figure 6. LIR motifs in MYO5A mediate its interaction with autophagosomes and are required for autophagosome–lysosome fusion

(a) Schematic representation of the MYO5A protein. The black boxes indicate the positions of LIR motifs mapped to the coiled-coil and globular tail domains. The positional information of the LIR motifs was analyzed using the iLIR web tool and is depicted below. On the right, the conservation of the LIR motifs across organisms is shown. (b) *MYO5A* KO cells were transfected with GFP-MYO5A, mEmerald-CC-GTD, or mEmerald-tagged P, Q, and DM LIR mutants in the CC-GTD domain of MYO5A, and co-transfected with mCh-LC3. Zoomed insets highlight co-localization events between mCh-LC3 and MYO5A constructs, as well as between mCh-LC3 and LAMP1. Scale bar, 1 μ m. (c–e) Quantification of co-localization events between LC3 and MYO5A (c), LAMP1 and MYO5A (d), and LC3 and LAMP1 (e) in *MYO5A* KO cells transfected with different LIR mutant constructs of MYO5A. Analysis was performed using the “Colocalization” plug-in of ImageJ. Data are plotted as mean \pm SEM from three independent experiments (N = 3, n = 60 cells). Statistical significance was assessed by one-way ANOVA with Kruskal–Wallis post-hoc test (**p < 0.01, ***p < 0.001, ns, non-significant). (f) *MYO5A* KO cells were transfected with mEmerald-CC-GTD or CC-GTD domains harbouring different LIR mutants (as indicated). Immunoprecipitation was performed using anti-GFP antibody, and lysates were probed for LC3 (N = 4). (g) Quantification of LC3 levels immunoprecipitated with GFP-tagged CC-GTD or LIR mutant constructs of MYO5A. Data are plotted as mean \pm SEM from three independent experiments (N = 4). Statistical significance was assessed by one-way ANOVA with Tukey post-hoc test.

Figure 7. Mean directional switches of autophagosomes are altered in the absence of MYO5A

(d) Time-lapse images of *MYO5A* KO cells co-transfected with mCh-LC3 and different LIR mutant constructs of MYO5A (as indicated). Scale bar, 1 μ m. (e) Quantification of mean directional changes of LC3-positive autophagosomes in *MYO5A* KO cells transfected with different LIR mutant constructs, analyzed using the “TrackMate” plug-in. Data are plotted as mean \pm SEM from three independent experiments (N = 3, n = 40 cells). Statistical significance was assessed by one-way ANOVA with Kruskal–Wallis post-hoc test (***p < 0.001, **p < 0.01, ns, non-significant).

7. Fig 6. When expressing the various Myo5 constructs were the cells deleted for the endogenous copy? If not, heterodimer formation could yield confusing results.

We thank the reviewer for raising this important point. All rescue experiments with the various MYO5A constructs were performed in the **MYO5A knockout background**, ensuring that no endogenous protein was present. This eliminates the possibility of heterodimer formation with endogenous MYO5A and confirms that the observed effects arise solely from the reintroduced constructs. To prevent any ambiguity, we have now **carefully revised the manuscript text, figure legends, and schematics to explicitly indicate this wherever relevant.**

References:

Evans, R.D. *et al.* (2014) “Myosin-Va and dynamic actin oppose microtubules to drive long-range organelle transport,” *Current Biology*, 24(15), pp. 1743–1750.

Klionsky, D.J. *et al.* (2021) “Guidelines for the use and interpretation of assays for monitoring autophagy (4th edition) ¹,” *Autophagy*, 17(1), pp. 1–382. Available at: <https://doi.org/10.1080/15548627.2020.1797280>.

Schott, D. *et al.* (1999) “The COOH-terminal domain of Myo2p, a yeast myosin V, has a direct role in secretory vesicle targeting,” *The Journal of cell biology*, 147(4), pp. 791–808.

Dear Ravi,

We have now received re-review reports from three referees, which I have included below. As you will see, you have addressed their concerns satisfactorily and I am happy to proceed towards publication; however, I would like you to address the remaining points of referees #1 and #3 in the discussion section. Before I can accept the manuscript, there are some remaining editorial points which need to be addressed. In this regard would you please:

- provide the manuscript ms as a .docx file, (not a PDF); with no 'track changes',
- label the corresponding author in the author list providing an email address on the title page,
- include up to five keywords,
- list references in alphabetical order without bullet points; for long author lists give the first 10 listed authors and then '+ et al.';
- remove DOI numbers from references,
- include a "Data availability" section,
- change the conflict of interests statement to a "Disclosure and competing interests statement",
- remove the AC/CrediT section from the text,
- list all figure callouts in the text sequentially and include a callout for Fig. 1B,
- submit a completed author checklist,
- upload each figure as an individual, high-resolution Figure file (not two or several figures uploaded as one PDF file as currently seen for Figure 4, Supplementary figure 3; Figure 7, Supplementary figure 5, Figure 8, supplementary figure 6, synopsis),
- rename Supplementary figure 1-6 as Figure EV1-EV6 (with the appropriate callouts in the main text),
- include a 'Reagents and Tools' section (there is a template in our guide to authors),
- provide source data (cc contact@embojournal.org),
- upload a synopsis image of 550 pixels wide by 200-400 high; you can also use something from the figures if that is easier,
- rename the "Materials and methods" section as "Methods", and
- name each section and use the following order: Title page - Abstract - Keywords - Introduction - Results - Discussion - Methods - Data Availability - Acknowledgements - Disclosure and Competing Interests Statement - References - Figure Legends - Table(s) - Expanded View Figure Legends.

We include a synopsis of the paper (see <http://emboj.embojournal.org/>). Please provide me with a general summary image, two sentence statement and 3-5 bullet points that capture the key findings of the paper.

I am looking forward to receiving your revised manuscript.

EMBO Press is an editorially independent publishing platform for the development of EMBO scientific publications.

Best wishes,

William

William Teale, PhD
Editor
The EMBO Journal
w.teale@embojournal.org

See also figure legend guidelines: <https://www.embojournal.org/page/journal/14602075/authorguide#figureformat>

- a point-by-point response to the referees' comments, with a detailed description of the changes made (as a word file).
- a word file of the manuscript text.
- individual production quality figure files (one file per figure)
- a complete author checklist, which you can download from our author guidelines

(<https://www.embopress.org/page/journal/14602075/authorguide>).

- Expanded View files (replacing Supplementary Information)

- a Reagents and Tools Table as part of the Methods section, which can be downloaded from our author guidelines

(<https://www.embopress.org/page/journal/14602075/authorguide#structuredmethods>)

We realize that it is difficult to revise to a specific deadline. In the interest of protecting the conceptual advance provided by the work, we recommend a revision within 3 months (6th Jan 2026). Please discuss the revision progress ahead of this time with the editor if you require more time to complete the revisions. Use the link below to submit your revision:

Referee #1:

This is a revised version of an original manuscript submitted for publication by Nambiar et al. a while ago, to which the authors introduced substantial new experimental data. In general, the authors characterize Myosin 5A as a regulator of late stages of autophagy, facilitating the targeting of mature autophagosomes to the lysosome. The authors utilized three different systems, including yeast, HeLa cells, and *C. elegans* neurons, to study the role of this motor in autophagy.

The original study suffered from a lack of mechanistic insights regarding the exact role of Myo5A in this pathway. The current study provides new data indicating that Myo5a acts both as a facilitator of autophagosome movement toward lysosomes and for recruiting the fusion machinery to the outer membrane of the autophagosome, a process mediated by one of its LIR motifs.

Overall, the study is significantly improved; however, a few issues should be further clarified to solidify the suggested model.

1. The authors start by showing the role of Myo2, the yeast orthology of Myo5a, in targeting autophagosomes to the vacuole. Next, the study moves to mammalian and *c. elegans* systems, focusing on Myo5a. One of the new mechanistic insights in the revised manuscript arises from the experimental data on LIR motifs, which are missing in the yeast. The authors must relate to this issue.

2. The images shown in Figure 5 are not always convincing. The authors are encouraged to test their images by TEM and possibly super-resolution fluorescent microscopy.

3. According to the suggested model, Myo5a, in addition to its association with autophagosomes via LC3 and the cytoskeleton, also recruits and stabilizes the fusion complex made of HOPS and SNAREs. How exactly is this achieved? Additionally, a more direct assessment of the interaction between Myo5a and the HOPS and/or the SNARE complexes is needed.

Referee #2:

The authors responded to all my concerns I raised for the original manuscript in a satisfactory manner.

Referee #3:

The revised manuscript has addressed my prior concerns and now presents a cohesive story that is well substantiated.

My only, very minor, suggestion is that on line 482 they introduce the hum2 mutants but never say what hum2 encodes. The reader must assume that it encodes a Myo5 homolog, but this should be clearly spelled out.

Also, it would be interesting to know if the yeast myo2 mutants affect binding to Atg8 or if they act by some other mechanism, but this is not critical.

All editorial and formatting issues were resolved by the authors.

Dear Prof. Manjithaya,

I am pleased to inform you that your manuscript has been accepted for publication in the EMBO Journal.

Congratulations to you and your team!

You may qualify for financial assistance for your publication charges - either via a Springer Nature fully open access agreement or an EMBO initiative. Check your eligibility: <https://link.springer.com/journal/44318/how-to-publish-with-us>

Yours sincerely,

William Teale

William Teale, PhD
Editor
The EMBO Journal
w.teale@embojournal.org

Please note that it is The EMBO Journal policy for the transcript of the editorial process (containing referee reports and your response letters) to be published as an online supplement to each paper. If you should prefer removal of any referee-only figures included in the point-by-point response(s), e.g. because they may still be used for future publication or because they have been reproduced from published work by others, please do let us know immediately via response email.

More information is available here: <https://link.springer.com/partners/embo-press/editorial-policies#Peer%20review>